# Parvalbumin interneurons regulate rehabilitation-induced functional recovery after stroke and identify a rehabilitation drug

Motor disability is a critical impairment in stroke patients. Rehabilitation has a limited effect on recovery; but there is no medical therapy for post-stroke recovery. The biological mechanisms of rehabilitation in the brain remain unknown. Here, using a photothrombotic stroke model in male mice, we demonstrate that rehabilitation after stroke selectively enhances synapse formation in presynaptic parvalbumin interneurons and postsynaptic neurons in the rostral forelimb motor area with axonal projections to the caudal fore-limb motor area where stroke was induced (stroke-projecting neuron). Reha-bilitation improves motor performance and neuronal functional connectivity, while inhibition of stroke-projecting neurons diminishes motor recovery. Stroke-projecting neurons show decreased dendritic spine density, reduced external synaptic inputs, and a lower proportion of parvalbumin synapse in the total GABAergic input. Parvalbumin interneurons regulate neuronal functional connectivity, and their activation during training is necessary for recovery. Furthermore, gamma oscillation, a parvalbumin-regulated rhythm, is increased with rehabilitation-induced recovery in animals after stroke and stroke patients. Pharmacological enhancement of parvalbumin interneuron function improves motor recovery after stroke, reproducing rehabilitation recovery. These findings identify brain circuits that mediate rehabilitation-recovery and the possibility for rational selection of pharmacological agents to deliver the first molecular-rehabilitation therapeutic.

Stroke is the leading cause of adult disability, with no approved medical therapy that promotes recovery. 70% to 80% of people who sustain a stroke have upper extremity impairment[1], and many of them do not regain functional use of the paretic arm, which can lead to difficulties in activities of daily living (ADLs) and engagement in com-munity life[2]. Unlike much of modern medicine, such as in cancer, immune or cardiac disease, there is no medicine for stroke recovery. Neurorehabilitation has the potential to improve recovery after stroke but in clinical practice has limited effect[3]. Rehabilitation has limited efficacy due to difficulties in access[1,4], intensity[5], comorbidities[6] and stroke severity[7]. Advancements in stroke recovery and rehabilitation have overall been modest, as evidenced by the limited effectiveness in large rehabilitation clinical trials[8]. To address these challenges, a comprehensive understanding of the circuits, physiology, and mole-cular systems that underly stroke rehabilitation is crucial[9].

Stroke recovery depends on the experience or surrounding envir-onment during recovery. Heightened perceptive stimuli, motor engagement, and social interaction promote functional recovery[10].

✉e-mail: Nokabe@mednet.ucla.edu; SCarmichael@mednet.ucla.edu

Rehabilitation is a therapy that enhances such restorative experiences through repetitive exposure to a stimulus, execution of motor training, and an enriched environment[10]. Stroke recovery and rehabilitation may improve outcomes through cellular mechanisms of neuroplasticity and over specific time points. Recovery after stroke is most robust in the weeks to months after the infarct, a period in which neurorehabilitative therapy is most effective[11]. Intensive task-specific training, such as repetitively using the affected forelimb, enhances recovery in human stroke and rodent stroke models during this period[11,12]. Task-specific neurorehabilitative training enhances synaptic plasticity in cortical neurons near the stroke site[13,14] and in corticospinal neurons near the stroke site that project to limb control areas of the spinal cord[15]. Previous studies have shown that spontaneous recovery after stroke occurs by employing biological and molecular mechanisms in the memory/learning system[16]. Motor rehabilitation is a re-learning process of acquired motor skills before the stroke. Thus, a molecular mechanism in motor learning might drive rehabilitation-induced motor recovery. A better understanding of the circuits and molecular mechanisms in rehabilitation could enable the development of molecular rehabilitation therapies. Such therapies would specifically activate neuronal circuits or molecules involved in rehabilitation-induced recovery, allowing a drug to mimic the effects of rehabilitation and reproduce functional recovery.

Motor learning and stroke recovery progress through excitatory and inhibitory network changes that influence synaptic plasticity[17,18]. Post-stroke plasticity changes are critically modulated by excitatory/inhibitory balance in the remaining neuronal circuits. Elevated excitatory activity, triggered by the blockade of inhibitory signals, enhances plasticity and facilitates functional recovery after stroke[19–21]. These findings suggest that the higher the excitatory signal relative to the inhibitory signal, the better recovery. However, this concept might be oversimplified because synaptic connections and circuits formed by excitatory and inhibitory neurons exhibit notable complexity[22,23]. A recent study has also shown that inhibition of a specific interneuron

subpopulation (VIP interneurons) enhances recovery after stroke, presumably through a disinhibition circuit mechanism[24]. Motor control is achieved by coordinated neuronal activity in many distinct neuronal populations[25,26]. In the normal brain, neuronal activity patterns are fine-tuned through a motor learning process mediated by synaptic plasticity[27]. Stroke causes disconnection of neural networks in adjacent and distant brain regions and disorganizes neuronal activity patterns necessary for motor control in a physiological state[28,29].

GABAergic inhibitory neuron diversity is necessitated by functional versatility in shaping the spatiotemporal dynamics of neural circuit operation[22], and fine-tuning of specific inhibitory circuits is required for efficient learning processes[30,31]. For example, motor skill learning with lever press tasks causes a decrease in the number of synapse terminals formed by somatostatin interneurons and increases those formed by parvalbumin interneurons[30]. These findings suggest that distinct changes in inhibitory circuitry among different interneuron subtypes may be crucial for facilitating functional recovery following stroke, particularly in tasks involving highly skilled movements. However, the distributed brain circuits that mediate neurorehabilitation-induced recovery after stroke have not been defined, their causal role in this process, and whether a specific pharmacological therapy for stroke recovery might be developed from these phenomena has not been assessed. Here, we show that neuronal circuits formed by presynaptic parvalbumin interneurons and stroke-projecting neurons mediate neurorehabilitation-induced recovery through network synchronization which provide molecular drug targets reproducing rehabilitation effects.

## Results

### Post stroke rehabilitation improves motor performance

To understand the neuronal circuit underlying functional recovery induced by rehabilitation, we developed a rehabilitation paradigm in the mouse that engages repetitive, skilled forelimb use by the affected limb, imitating rehabilitation for human stroke patients[12,32] (Fig. 1a and

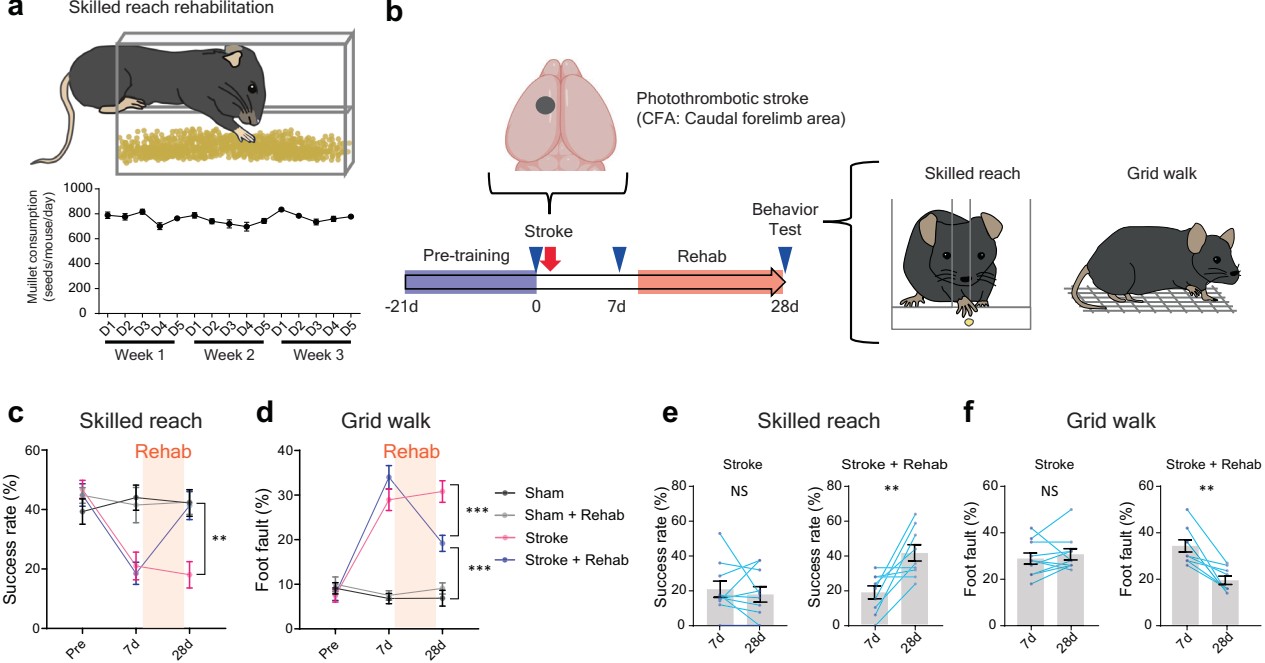

**Fig. 1 | Rehabilitative training recovers motor performance after stroke.**
**a** Rehabilitation box used in the study. Mice stably engage in reach-to-grasp training for 3 weeks, 5 days a week during recovery. n = 12. The error bars are smaller than the symbols. **b** Timeline of rehabilitation and behavioral testing. The blue triangles indicate behavioral tests. Pretraining for behavior tests and rehabilitation periods are indicated as blue and red rectangles. **c, d** Skilled reaching test

(**c**): success rate (time by group, F (6, 64) = 10.35, P < 0.0001) and grid walk test (**d**): foot fault rate (time by group, F (6, 64) = 26.49, P < 0.0001). Two-way repeated measure ANOVA with Sidak's multiple comparison test. **P < 0.01, ***P < 0.001. **e, f** Functional improvement in the recovery period in the skilled reaching test (**e**) and the grid walk test (**f**). Two-tailed paired t-test. (**a**–**f**) n = 8 (Sham), 8 (Sham + Rehab), 10 (Stroke) or 9 (Stroke + Rehab). (**b**) Created in BioRender[127].

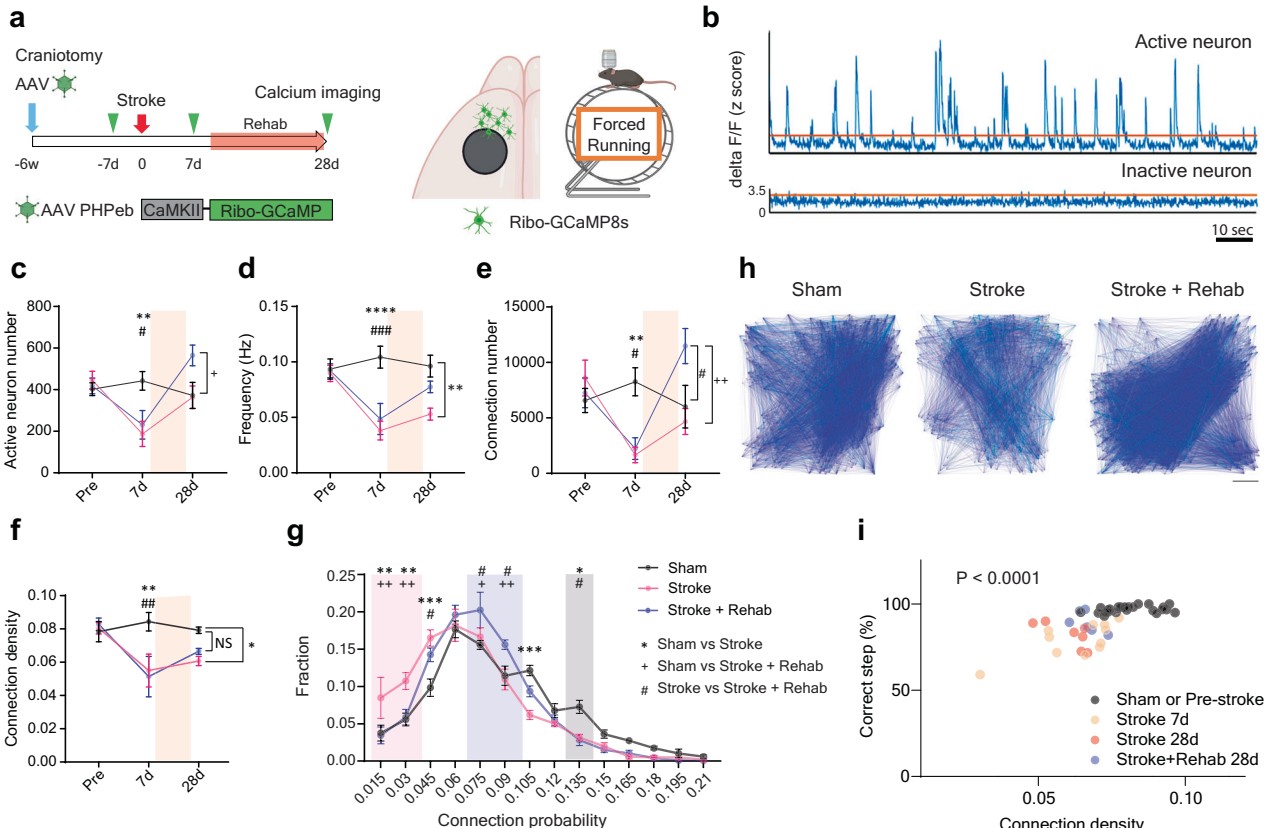

**Fig. 2 | Rehabilitative training recovers functional connectivity after stroke.**
**a** Timeline of calcium imaging study. The green triangles indicate calcium imaging sessions. **b** Representative trace of calcium transient obtained from active (upper) and inactive (lower) neurons. **c** Active neuron number (time by group, F (4, 28) = 5.504, P = 0.0021). **d** calcium transient event frequency (time by group, F (4, 28) = 3.850, P = 0.0129). **e** functional connection number (time by group, F (4, 28) = 8.518, P = 0.0001). **f** connection density (time by group, F (4, 26) = 2.936, P = 0.0397). Mixed-effects model, **P < 0.01, ****P < 0.0001: Sham (n = 4) vs Stroke (n = 7), # P < 0.05, ## P < 0.01, ### P < 0.001: Sham vs Stroke + Rehab (n = 6), + P < 0.05, ++ P < 0.01: Stroke vs Stroke + Rehab, Tukey's multiple comparisons test. Two-sided. Tukey's HSD correction for multiple comparison. **g** Fractional

distribution of connection probability in individual neurons. Red, blue, and black areas indicate the ranges where the fraction increases in stroke, stroke + rehab, and sham groups. Two-way ANOVA (F (26, 196) = 3.789, P < 0.0001), *P < 0.05, **P < 0.01, ***P < 0.001: Sham vs Stroke, # P < 0.05: Sham vs Stroke + Rehab, + P < 0.05, ++ P < 0.01: Stroke vs Stroke + Rehab, Tukey's multiple comparisons test. (**c–g**) n = 4 (Sham), 7 (Stroke) or 6 (Stroke + Rehab). **h** Representative connection maps 28 days after the stroke. Scale bar, 100 µm. **i** Correlation between motor performance in the grid walking and the connection density. Pearson correlation (n = 49, r = 0.671, P < 0.0001). Two-sided. Data are presented as means ± sem. (**a**) Created in BioRender[127].

Supplementary Fig. 1). Such rehabilitation enables intense reach-to-grasp training (> 800 reaches/day). We tested the effect of this rehabilitation paradigm on functional recovery after stroke in primary motor cortex (M1, caudal forelimb area, CFA, Supplementary Fig. 2) using two motor tests, the skilled forelimb reaching test and the grid walk test, which assess motor performance in reach-to-grasp behavior and innate walking precision, (Fig. 1b). We found that stroke significantly decreased the success rate of skilled reaching (Fig. 1c) and diminished gait function (Fig. 1d). Skilled reach rehabilitation completely recovers the motor performance in the skilled reaching task (Fig. 1c, e) and also improved functional recovery in gait (grid walk) (Fig. 1d,f), indicating that intensive rehabilitative therapy promotes motor recovery at or near non-stroke motor performance.

Motor recovery after stroke is associated with plasticity in brain areas anatomically connected to the stroke site. The ipsilesional premotor cortex and the isotopic area to the stroke site in the contralesional hemisphere are regions well-known for contributing to motor recovery[33,34]. To identify the brain area most important to motor recovery, we tested the functional significance of the rostral forelimb area (RFA) within ipsilesional premotor cortex and the contralesional CFA by inducing a second stroke in these sites after successful rehabilitation (Supplementary Fig. 3a). A second stroke in ipsilesional RFA eliminated rehabilitation-induced behavioral gains in both tasks, while

a contralesional stroke did not cause significant changes (Supplementary Fig. 3b, e). These results indicate that intracortical circuits in RFA (premotor cortex) are a site of rehabilitation-induced recovery.

## Post stroke rehabilitation restores functional connectivity in specific circuits

Execution of precise motor tasks requires neuronal population activity in the motor and premotor cortex[35,36]. Stroke causes not only neuronal death in the infarct core but also cortical network uncoupling characterized by less synchronized firing activity. This uncoupling, quantified as decreased functional connectivity, is closely associated with behavioral deficits[28,37,38]. We assessed neuronal activity and functional connectivity in the main neurons with intracortical connections (layer 2/3 neurons) in the RFA, employing a two-photon microscope equipped with a rotating grid wheel (Fig. 2a, b). Previous studies demonstrated that walking behavior on a rotating grid sensitively detects motor deficits and network uncoupling after a stroke[28,39]. Rehabilitation-induced stroke recovery is present in both skilled reach and grid-walking tests. However, the rehabilitation activity in these studies is repetitive skilled reach. Using the same behavioral paradigm for training and assessment could introduce confounding factors, as post-stroke training can lead to compensatory movements that affect outcomes[40–42]. Therefore, we selected the grid walk paradigm over the

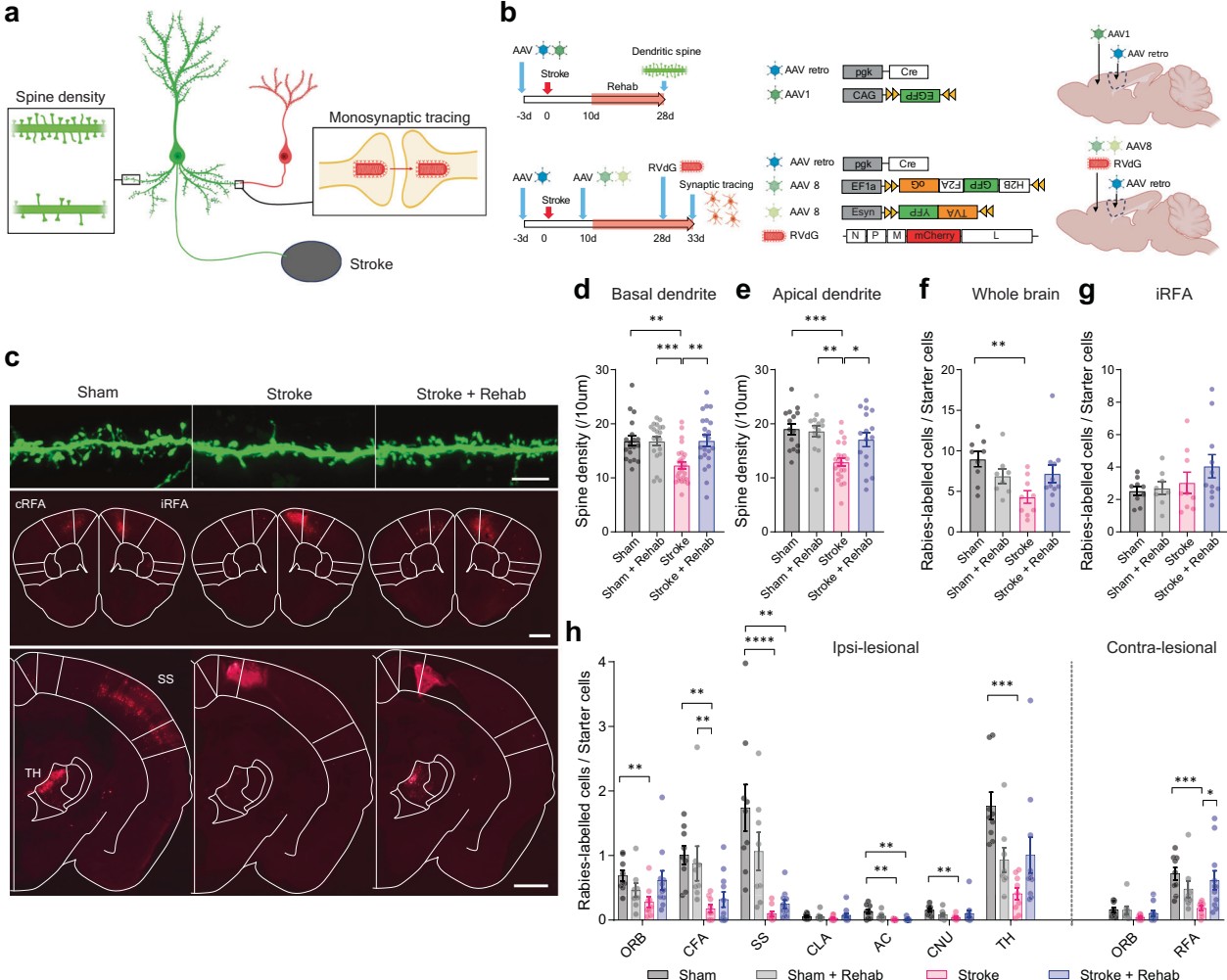

**Fig. 3 | Rehabilitation restores synaptic input to stroke-projecting neurons.**
**a** Approach for synaptic inputs in stroke-projecting neurons. **b** Timeline, virus vectors, and virus injection locations in dendritic spine analysis (upper) and monosynaptic tracing (lower). The dashed lines in the brain illustration indicate future stroke sites (CFA). **c** Representative images showing dendritic spines (upper: scale bar 5 μm) and G-deleted rabies virus (RVdG) labeled cells (middle and lower: scale bar 1 mm). cRFA: contralesional rostral forelimb area, iRFA: ipsilesional RFA, TH: thalamus, SS: somatosensory area. **d**, **e** Spine density in layer 5 stroke-projecting neurons; basal dendrite (**d**), n = 19 (Sham), 20 (Sham + Rehab), 27

(Stroke) or 22 (Stroke + Rehab), apical dendrite (**e**), n = 15 (Sham), 15 (Sham + Rehab), 22 (Stroke) or 16 (Stroke + Rehab). Kruskal-Wallis test. 4–5 animals per group. **f**–**h** The number of RVdG-labeled cells normalized by the starter cells in the whole brain (**f**), RVdG injection neighbor: RFA (**g**), and distant brain areas (**h**). ORB: orbital area, CFA: caudal forelimb area, SS: somatosensory area, CLA: claustrum, AC: anterior cingulate area, CNU: cerebrum nuclei (striatum and pallidum), TH: thalamus. Kruskal-Wallis test. n = 9 (Sham), 8 (Sham + Rehab), 9 (Stroke) or 11 (Stroke + Rehab). (**a**, **b**) Created in BioRender[127].

reaching task for the calcium imaging studies. Stroke reduced the number of active neurons, calcium transient frequency, connection number, and connection density in these neurons, though they are distant from the infarct (Fig. 2c–f, the distance from the infarct to the center of the imaging field is 925 ± 25 μm. n = 6). This reduction signifies a deterioration in neuronal activity and functional connectivity. While the number of active neurons spontaneously increased over time, both the frequency and connection density remained depressed 28 days post-stroke (Fig. 2d, f). Rehabilitation yielded a significant increase in the number of active neurons and functional connections. Rehabilitated animals showed no significant decrease in frequency or connection density compared to sham animals. To further characterize the functional connectivity changes, we calculated the connection probability of a single neuron as the ratio of functional connections to the theoretical maximum connection number in the population. We found that rehabilitation reduced the fraction of neurons exhibiting sparse connectivity in RFA and increased the neurons with higher connection probability compared to non-treated stroke animals (Fig. 2g). These results suggest that rehabilitation enhances functional

connectivity in severely affected neurons. Our data also revealed a significant correlation between connection density and motor performance in the grid waking (Fig. 2i).

## Post stroke rehabilitation restores synaptic inputs to stroke-projecting neurons

The significant neuronal activity and functional connectivity changes after post-stroke rehabilitation suggest synaptic alterations in the RFA as a direct effect of stroke, and as a substrate for rehabilitation. We next investigated whether anatomical connectivity also changes in the RFA with rehabilitation. In both healthy and stroke animals, skilled reach training induces formation of new dendritic spines that serve as the primary sites of synaptic connections[14,43]. We measured dendritic spine density in the RFA and further identified the source of synaptic inputs by monosynaptic rabies virus tracing. Using an intersectional virus approach, we targeted two types of cortical projection neurons implicated in motor control after stroke: corticospinal neurons (Supplementary Fig. 4a) and stroke-projecting neurons (Fig. 3a, b). Corticospinal neurons represent the direct cortical outputs to the spinal

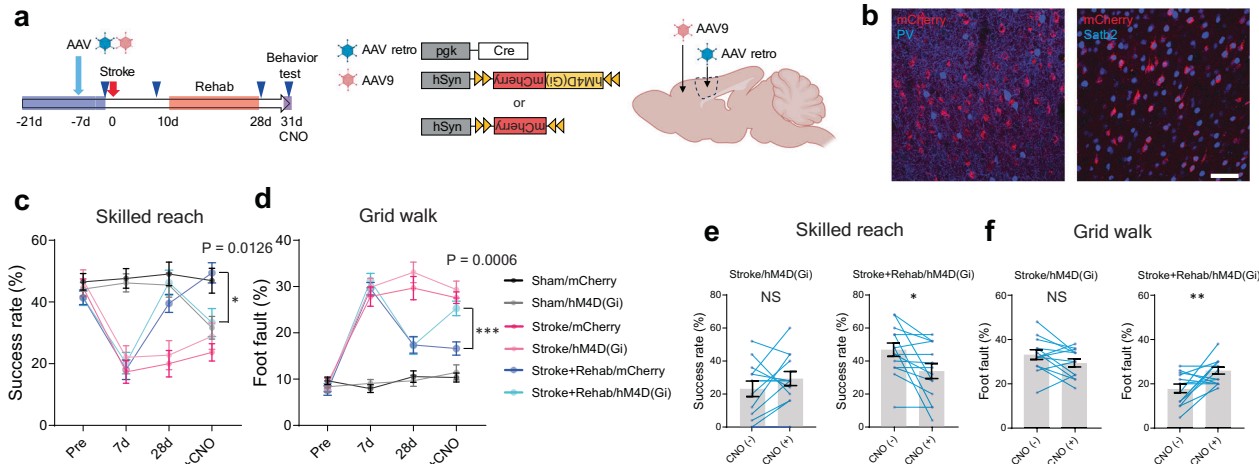

**Fig. 4 | Stroke-projecting neurons are necessary for rehab-induced functional recovery. a** Timeline, virus vectors, and injection sites in the chemogenetic inhibition targeting stroke-projecting neurons. **b** Representative images of the stroke-projecting neurons with hM4D(Gi)-mCherry in the RFA layer 5. The stroke-projecting neurons express a cortical projection neuron marker, Satb2 (right), but not PV (left). scale bar 50 μm. Similar staining was confirmed in the sections from 9 mice. See supplemental material for the quantification. **c, d** Motor performance in the skilled reaching test (**c** time by group, F (15, 204) = 9.930, P < 0.0001) and the grid walk test (**d** time by group, F (15, 204) = 17.86, P < 0.0001). Two-way repeated measure ANOVA, Sidak's multiple comparison test. **e, f** Motor performance changes induced by the chemogenetic inhibition in the skilled reaching test (**e**) and the grid walk test (**f**). Two-tailed paired t-test. (**c–f**) n = 11 (Sham/mCherry), 11 (Sham/hM4D), 12 (Stroke/mCherry), 13 (Stroke/hM4D), 13 (Stroke+Rehab/mCherry) or 14 (Stroke+Rehab/hM4D) Data are presented as means ± sem. *P < 0.05, **P < 0.01, ***P < 0.001, ****P < 0.0001. (**a**) Created in BioRender[127].

cord and play a pivotal role in motor control[44]; the integrity of the corticospinal tract is a strong predictor of stroke outcomes[45]. Conversely, stroke-projecting neurons are defined by their axonal projection to the stroke site, detected by retrograde labeling from the future stroke site (retrograde AAV injection onto the CFA before the stroke in this study). Stroke-projecting neurons lose their projection target and receive retrograde injury signals, which impact neuronal excitability, morphology, and gene expression[46], and may also possess unique plasticity. Before conducting synaptic measurements, we labeled these neuron types by injecting retrograde AAV expressing GFP and tdTomato into the M1 and the cervical spinal cord (C7) to determine whether corticospinal neurons and stroke-projecting neurons (M1-projecting neurons) are distinct. We found that whereas both neuron types were predominantly localized in layer 5 in the RFA, these neurons were rarely co-labeled (Supplementary Fig. 4b), suggesting corticospinal and stroke-projecting neurons are anatomically distinct. Then, we confirmed the specificity of the rabies virus labeling by injecting helper AAVs and rabies virus in the absence of Cre. We found that the rabies virus causes little leak and labeling in the injection site and no label in the distance brain areas without Cre, indicating minimum leakage by non-specific labeling (Supplementary Fig. 5). In dendritic spine analysis, we observed that neither stroke or rehabilitation affected dendritic spine density in corticospinal neurons in the RFA (Supplementary Fig. 4c, d). In corticospinal neurons, rabies tracing showed decreased synaptic inputs from the CFA (near the stroke site) and the somatosensory cortex; rehabilitation did not cause significant changes in any synaptic input sources (Supplementary Fig. 4e–g). In contrast, stroke-projecting neurons lost a substantial proportion of dendritic spines after stroke in both layers 2/3 and 5 (Fig. 3c–e and Supplementary Fig. 6). Monosynaptic tracing revealed that synaptic inputs were lost in major brain areas projecting to the stroke-projecting neurons (Fig. 3f–h and Supplementary Fig. 7). Rehabilitation restored dendritic spine density and a portion of synaptic inputs to the stroke-projecting neurons from some brain areas including the contralesional RFA, peri-infarct CFA, and thalamus. (Fig. 3d–h and Supplementary Figs. 6, 7). These data identify stroke-projecting neurons in the RFA as a population uniquely affected by loss of synaptic inputs after stroke and uniquely responsive in their input connections to rehabilitation, in a brain region integral to rehabilitation-induced recovery.

## Stroke-projecting neurons mediate motor performance

The restoration of synaptic inputs to stroke-projection neurons after rehabilitation suggests a functional role in motor recovery. To determine if stroke-projecting neurons are causally involved in rehabilitation-induced recovery, we manipulated the stroke-projecting neurons with designer receptors exclusively activated by designer drugs (DREADDs) after rehabilitation (Fig. 4a, b). The inhibitory DREADD receptor, hM4D(Gi) or mCherry as a non-DREADD control were virally transfected in stroke-projecting neurons in the ipsilesional RFA before the stroke (Fig. 4a). Immunohistochemical analysis (Fig. 4b and Supplementary Fig. 8) revealed that virus expression co-localized with a cortical projecting neuron marker, Satb2 (86.1 ± 1.5%), while showing rare co-localization with an interneuronal marker, parvalbumin (0.38 ± 0.08%). Parvalbumin (PV) interneurons primarily form local cortical connections, indicating effective retrograde labeling without diffusive viral infection. The layer distribution analysis revealed that 30.4 ± 1.3% and 61.1 ± 1.5% of stroke-projecting neurons localized in layers 2–3 and 5, respectively (Supplementary Fig. 8). We also found that their axons densely projected to the contralateral cortex, striatum, thalamus, and cerebral peduncle, but the peri-infarct projection was sparse, indicating thorough destruction of the axonal projection to the CFA and limited axon regeneration (Supplementary Fig. 8). We assessed motor recovery 28 days after stroke and tested stroke-projecting neuron inhibition 3 days later. We assessed motor performance 15 min after the injection of the chemogenetic ligand, clozapine N-oxide (CNO). Whereas neither hM4D(Gi) or mCherry expression affects baseline motor performance or recovery (Fig. 4c, d: Pre, 7d and 28d), chemogenetic inhibition of stroke-projecting neurons significantly deteriorates motor performance in sham control and rehabilitation-stroke animals in the skilled reaching test (Fig. 4c, e: +CNO). Chemogenetic inhibition also affects the motor performance of the rehabilitation-treated animals in the grid walk test (Fig. 4d, f). These results indicate that stroke-projecting neurons in rehabilitation animals play a significant role in motor recovery.

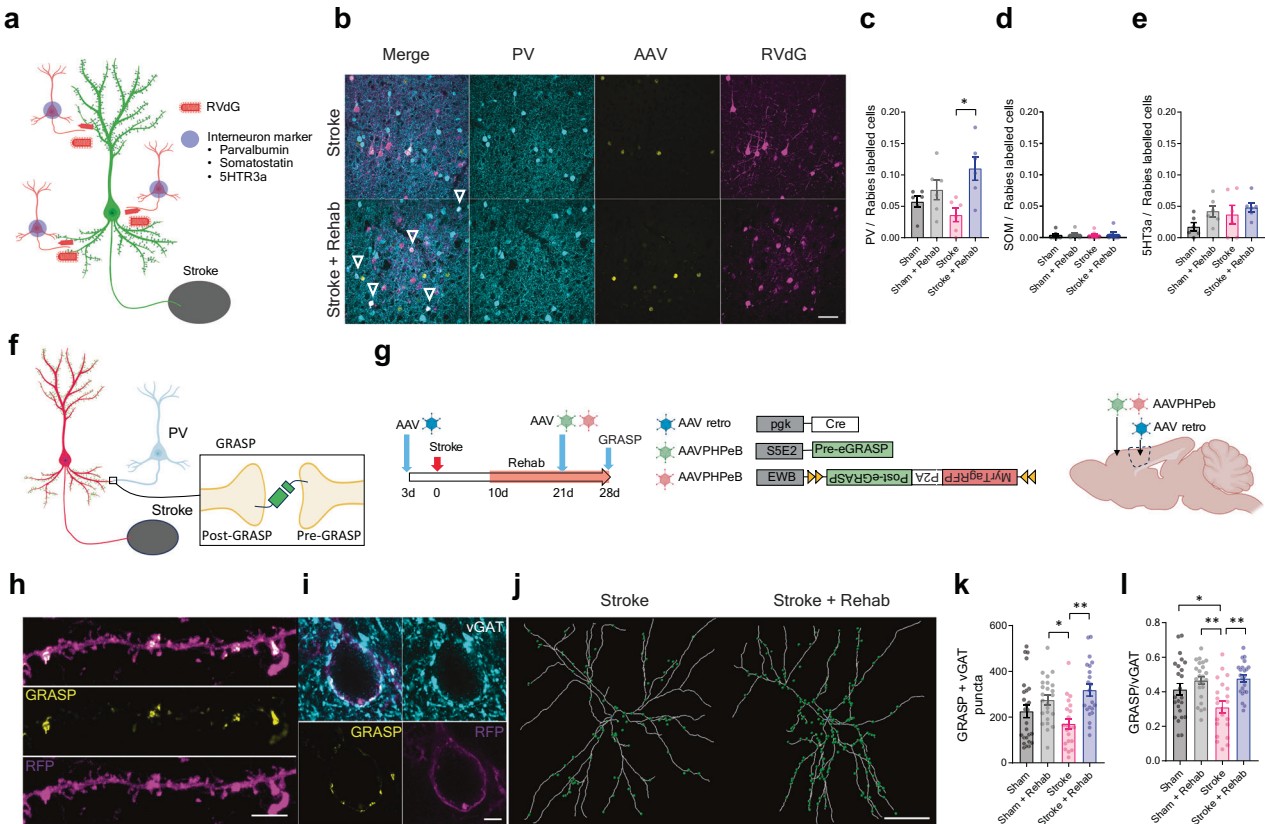

**Fig. 5 | Rehabilitation induces synapse formation from PV interneurons to stroke-projection neurons. a** Schematic illustration of interneuron marker labeling combined with monosynaptic tracing. PV: parvalbumin, SOM: somatostatin, 5HT3a: serotonin receptor 3a. **b** Representative images of RVdG labeled PV interneurons. The green channel shows the GFP labeled starter cells. Scale bar 50 μm **c–e** The ratio of PV (**c**), SOM (**d**), 5HT3a (**e**) in total local inputs. Kruskal-Wallis test, n = 6. **f** Schematic illustration of GRASP labeling of synapse formed by stroke-projecting neuron and PV interneuron. **g** Timeline, virus vectors, and injection sites

in the GRASP study. **h, i** Representative images of GRASP labeled dendrite (H, scale bar 5 μm) and soma (I, scale bar 10 μm). **j** Example of reconstruction of dendritic tree and GRASP+vGAT puncta. Scale bar 30 μm. GRASP expression patterns were consistent across the staining sessions and the animals **k, l** The number of GRASP +vGAT puncta (**k**) and the proportion of GRASP positive synapse in the total vGAT positive synapse. (**l**) on the stroke projecting neurons. Kruskal-Wallis test, n = 26 (Sham), 26 (Sham + Rehab), 23 (Stroke) or 24 (Stroke + Rehab). (**a, f, g**) Created in BioRender[127].

## Rehabilitation selectively enhances synapse formation from parvalbumin interneurons to stroke-projecting neurons

Temporal patterns of neuronal activity are regulated by both local cortical dynamics and external inputs from distant areas. In addition to external inputs to stroke-projecting neurons, local connections in inhibitory neurons may mediate circuit connectivity and dynamics through mechanisms of synaptic plasticity[47]. Thus, we next examined the composition of local inhibitory synaptic inputs to stroke-projecting neurons by immunohistochemical labeling combined with monosynaptic rabies tracing (Fig. 5a). We found that rehabilitation significantly increased synaptic input from PV interneurons to stroke-projecting neurons (Fig. 5b, c), but not from the other major classes of inhibitory neurons: somatostatin, or 5HT3a interneurons (Fig. 5d, e). Unlike stroke-projecting neurons, this selective synaptic change was not observed in corticospinal neurons after stroke (Supplementary Fig. 4h–j). Importantly, the density of cortical interneurons did not change by stroke or rehabilitation (Supplementary Fig. 9a–e), and increased PV interneuron inputs were not correlated with PV interneuron density (Supplementary Fig. 9f). Because rabies virus labeling could be affected by virus toxicity and neuronal activity, we confirmed the rehabilitation-induced PV synapse formation using a synapse-specific labeling technique, GFP Reconstitution Across Synaptic Partner[48] (GRASP: Fig. 5f-i and Supplementary Fig. 10). GRASP utilizes split GFP, with pre- and post-synaptic components (pre- and post-GRASP), which emit a green fluorescent signal exclusively when expressed at both pre- and post-synaptic terminals. To specifically

detect synapses formed by presynaptic PV interneuron and postsynaptic stroke-projecting neuron, we induced presynaptic GRASP in PV interneurons using the PV interneuron-specific enhancer, S5E2[49] (PV specificity: 91.3 ± 1.64%) and postsynaptic GRASP in the stroke-projecting neurons using retrograde AAV-Cre vector. Consistent with the findings of the monosynaptic tracing study, we noted a marked increase in PV synapses on stroke-projecting neurons in stroke animals treated with rehabilitation (Fig. 5j, k, and Supplementary Fig. 10e). This increase in PV synapses was not accompanied by changes in total GABAergic synapse inputs (vGAT, Supplementary Fig. 10d). Interestingly, stroke significantly decreased the proportion of PV synapses within the total GABAergic synapse inputs and rehabilitation restored it (Fig. 5l). These data indicate that rehabilitation selectively increases PV interneuron input to stroke-projecting neurons and adjust the composition of GABAergic synapse inputs.

## Parvalbumin interneurons mediate functional recovery

Given the selective synapse formation from PV interneurons to stroke-projecting neurons in rehabilitation, we tested whether post-stroke rehabilitation would activate this neuronal circuit. To this end, we assessed whether post stroke rehabilitation induces activity-dependent gene expression and plasticity changes in the PV/stroke-projecting neuron circuit in RFA by the detection of immediate early genes Zif268 and FosB, and of perineuronal nets, a determinant of PV neuron plasticity[50] (Supplementary Fig. 11a, g). We observed that rehabilitation significantly increased Zif268 expression in both types

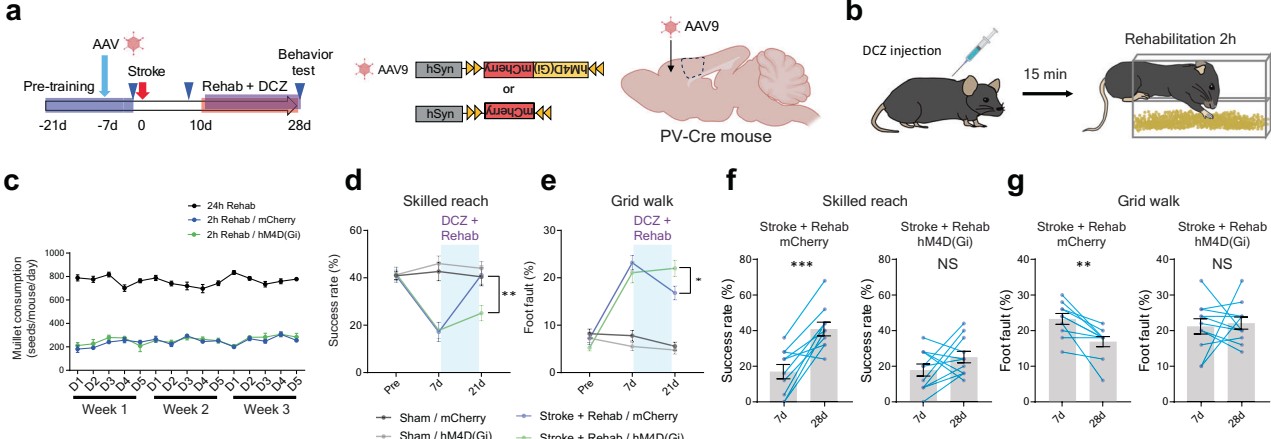

**Fig. 6 | PV interneuron activation is necessary for recovery. a** Timeline, virus vectors, and injection sites in the chronic chemogenetic inhibition targeting PV interneurons. **b** Procedure for rehabilitation with chemogenetic inhibition. **c** Millet seed consumption with 2-h rehabilitation. n = 6. **d, e** Motor performance in the skilled reaching test (**d** time by group, F (6, 68) = 7.887, P < 0.0001) and the grid walk test (**e** time by group, F (6, 68) = 19.69, P < 0.0001). Two-way repeated measure ANOVA: Stroke + Rehab/mCherry vs Stroke + Rehab/hM4D(Gi), Sidak's multiple comparison test. **f, g** Functional recovery by rehabilitation in the skilled reaching test (**f**) and the grid walk test (**g**). Two-tailed paired t-test. (**d–g**) n = 9 (Sham/mCherry), 8 (Sham/hM4D), 10 (Stroke+Rehab/mCherry) or 11 (Stroke +Rehab/hM4D). All data are presented as means ± sem. *P < 0.05, **P < 0.01, ***P < 0.001. (**a, b**) Created in BioRender[127].

of neurons (Supplementary Fig. 11c, d) and FosB expression in the stroke-projecting neurons (Supplementary Fig. 11e,f). Rehabilitation also decreased the ratio of PV interneurons surrounded by perineuronal nets (Supplementary Fig. 11i,j). These data indicate activation and enhanced plasticity of these neuronal circuits by rehabilitation.

We further tested whether the activation of these neuronal circuits is necessary for functional recovery by rehabilitation. We induced inhibiting hM4D(Gi) or mCherry control in either stroke-projecting neurons or PV interneurons using retrograde AAV-Cre (Supplementary Fig. 12a) and a PV-Cre mouse line (Fig. 6a). We injected the chemogenetic ligand, DCZ, 15 min before every rehabilitation session through the recovery period (Fig. 6b). We validated the efficacy of DREADD inhibition through the treatment period using 2 photon microscope calcium imaging in PV-Cre mice injected with Cre-dependent AAVs expressing hM4D(Gi) and GCaMP8s (Supplementary Fig. 13). We found this DREADD protocol significantly inhibited the movement-induced PV interneuron activation at 10 and 28 days after the stroke (Supplementary Fig. 13). We limited access to the rehabilitation apparatus for 2 h during which DCZ exerts maximum effects[51]. Compared to the previous studies, this shorter training period decreased the total reaches (millet seed consumption) but remained effective in improving motor performance after stroke (Fig. 6c–e and Supplementary Fig. 12c–e: mCherry). While the inhibition of stroke-projecting neurons influenced rehabilitation-induced motor recovery exclusively in the grid walk task (Supplementary Fig. 12f,g), inhibiting PV interneurons had a significant and broader impact, diminishing the effectiveness of recovery in both skilled reaching and the grid walk (Fig. 6f,g). This indicates that PV interneuron activation has an essential role in rehabilitation-induced recovery in both reach-to-grasp and precision gait function.

## Stroke decreases the fraction of fast-rising large amplitude IPSCs, and rehabilitation restores it

Our histological analysis revealed that stroke-projecting neurons undergo structural synaptic alterations following stroke and rehabilitation. To investigate whether these neurons also exhibit changes in synaptic inputs, we conducted patch-clamp recordings of stroke-projecting neurons. We labeled the stroke-projecting neurons with retrograde AAV expressing tdTomato and recorded spontaneous EPSC/IPSC in the tdTomato positive stroke-projecting neurons (CFA-projecting neurons in Sham animals) in the RFA brain slices. We also validated external synaptic inputs by recording evoked EPSCs in response to optogenetic stimulation of thalamocortical axons (Fig. 7a).

In the EPSC recordings, stroke induced a significant increase in peak EPSC amplitude (Fig. 7d), while EPSC frequency remained unchanged (Fig. 7c). In contrast, rehabilitation significantly increased EPSC frequency compared to stroke animals (Fig. 7c). In optogenetic recordings, stroke reduced the EPSC probability to optogenetic stimulation of thalamic afferents (Fig. 7f) without significant differences in peak amplitude (Fig. 7g), suggesting that excitation of thalamic afferents to the CFA produces less reliable responses in stroke-projecting neurons in both stroke and rehabilitation animals. Our histological data, including dendritic spine analysis and rabies virus tracing (e.g., increased spine density in rehabilitation animals compared to untreated stroke animals and reduced synaptic input from distant brain areas), are primarily consistent with EPSC frequency/probability data but not EPSC amplitude. These findings suggest that EPSC frequency/probability is more closely related to structural synaptic alterations (e.g., synapse loss or formation), whereas EPSC amplitude is influenced by additional factors such as intrinsic excitability, postsynaptic receptor expression, presynaptic vesicle release probability, and phasic/tonic inhibitory inputs mediated by transcriptional regulation. (see Discussion for details).

In the IPSC recordings, we found that stroke increased the IPSC frequency (Fig. 7i) without changing the IPSC peak amplitude (Fig. 7j), indicating increased inhibitory synaptic inputs after stroke. Conversely, rehabilitation showed no significant changes. In the GRASP study, stroke decreases the fraction of PV interneuron synapses without total inhibitory synapse number, and rehabilitation restores these abnormalities. Thus, stroke-projecting neurons may also show fractional changes in IPSCs. We further analyzed the IPSC data to determine whether our IPSC recordings capture PV interneuron functionality. Since PV interneurons target the α1 subunit-containing GABAA receptors, they predominantly generate fast-rising IPSCs[52–54]. Also, the somatic location of PV interneuron inputs results in large-amplitude IPSCs. We have previously shown that the loss of fast-rising large amplitude IPSCs in a mouse model of Alzheimer's disease mirrors the functional and physical loss of PV interneurons[55]. Using as thresholds the mean rate of rise and peak amplitude of all recorded IPSCs, we estimated the fraction of large amplitude fast-rising events in each group, most likely originating from PV

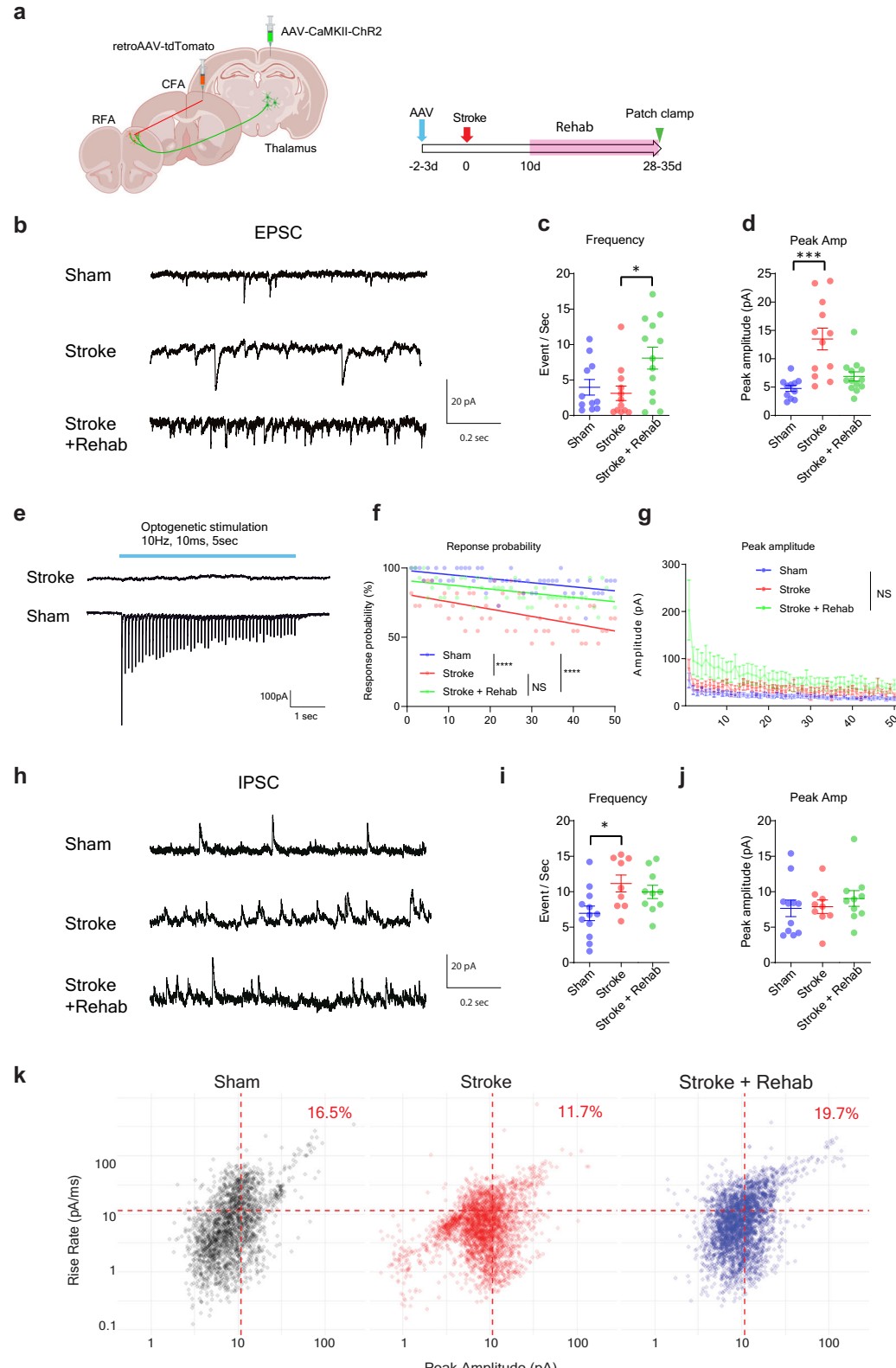

interneuron inputs. The data show a reduction in the fraction of large amplitude fast rate of rise IPSCs following stroke, with recovery observed after rehabilitation (Fig. 7k), indicating that stroke reduces, and rehabilitation recovers fractional synaptic inputs from PV interneurons. Additionally, these findings are consistent with the idea that the increased IPSC frequency observed in stroke animals is primarily attributable to small-amplitude or slow-rising inhibitory synaptic inputs from interneurons other than PV interneurons.

## Parvalbumin interneurons regulate functional connectivity after stroke

To study how PV interneurons might enhance functional recovery after stroke, we next asked whether PV interneurons could modulate

**Fig. 7 | Stroke reduces the fraction of fast-rising large amplitude IPSCs and rehabilitation restores it. a** Virus injection and time course for patch-clamp recording. **b** Representative spontaneous EPSC traces. **c, d** Frequency (**c**), and peak amplitude (**d**) of spontaneous EPSC. Kruskal-Wallis test, n = 11 (Sham), 12 (Stroke), or 13 (Stroke + Rehab). **e** representative traces of EPSCs evoked by optogenetic stimulation targeting thalamic axons. **f, g** Response probability (**f**), and peak amplitudes (**g**) responding to 50 sequential optogenetic stimulations. Generalized linear mixed model (Fixed effect: Stimulation number; *P* < 0.0001, Group; *P* < 0.0001). Two-sided. Tukey's HSD correction for multiple comparison. n = 11 (Sham), 12 (Stroke), or 13 (Stroke + Rehab). **h** Representative spontaneous IPSC traces. **i, k** Frequency (**i**), and peak amplitude (**j**) of spontaneous IPSC. Kruskal-Wallis test, n = 12 (Sham), 9 (Stroke), or 10 (Stroke + Rehab). *$P$ < 0.05, ***$P$ < 0.001,****$P$ < 0.0001. **k** Distribution of IPSC rise rate and peak amplitude. The red dashed lines indicate the mean peak amplitude and rate of rise of all IPSC events from all recordings. The large amplitude with fast rate of rise events above both mean values (upper right quadrant) amount to 16.5%, 11.7%, and 19.7% of all IPSC events in the Sham, Stroke, and Stroke + Rehab groups, respectively. Chi-squared test; X-squared = 73.4, df = 2, $P$ = 2.2e-16, Bonferroni post hoc test, two-sided; Sham vs Stroke: $P$ < 0.0001, Sham vs Stroke + Rehab: $P$ = 0.0092, Stroke vs Stroke + Rehab; $P$ < 0.0001. Data are presented as means ± sem. (**a**) Created in BioRender[127].

neuronal connectivity associated with motor control. Previous studies had shown that PV interneurons regulate synchronous neuronal activity in the perception of sensory information[56,57]. To gain insights into the causal role of PV interneurons in motor function, we conducted a chemogenetic inhibition study combined with calcium imaging. The hM4D(Gi) receptor was induced in PV interneurons and neuronal activity was recorded in the excitatory neurons in RFA (Fig. 8a). Given the distinct roles of PV interneurons depending on their activity state[56,57], we recorded neuronal activity in both stationary (stop epoch) and active (running epoch) states in voluntary running on a grid wheel (Fig. 8b). Similar to the previous forced running experiment, the stroke animals exhibited severe motor deficits (Fig. 8d) accompanied by a persistent reduction in connection density in both running and stop epochs Supplementary Fig. 14c-j). Four weeks after the stroke, we assessed the effects of inhibiting PV interneurons on motor performance and neuronal activity. We found that inhibition of PV interneurons impaired motor performance during grid running in sham and rehabilitation animals but had no effect on stroke animals (Fig. 8d, e). Notably, inhibition of PV interneurons also decreased functional connectivity during the running epoch in all groups (Fig. 8f, g). Consistent with the correlation between motor function and functional connectivity in the previous calcium imaging (Fig. 2i), the changes in motor performance and functional connectivity showed a significant correlation (Fig. 8h). As expected from the known PV interneuron functions of feedback and feedforward inhibition to excitatory neurons, chemogenetic inhibition of PV interneurons resulted in a significant increase in active neuron count and calcium transient frequency during the stop epoch (Supplementary Fig. 15). Surprisingly, we observed a dramatic increase in functional connectivity during the stop epoch in stroke animals, while sham animals exhibited decreased functional connectivity (Fig. 8i, j). Collectively, these results indicate PV interneuron inhibition decreases the connection density associated with motor impairment in normal animals but induces distinctive effects in stroke animals. This implies that PV interneurons are involved in the regulation of neural network connections, and stroke disrupts circuit control mediated by PV interneurons.

## Gamma power changes after stroke in the mouse stroke model and human patients

The precise timing of neuronal activity and population synchrony are coordinated by network oscillations[58]. PV interneurons play a pivotal role as key cellular elements in the generation of such network oscillations, especially gamma waves, which establish oscillatory envelopes of increased neuronal activity onto neuronal networks[59]. To determine whether neuronal oscillations are associated with stroke recovery, we assessed network oscillation in the premotor cortex/RFA of stroke animals. Recording electrodes were implanted in the RFA, and a transparent acrylic column was placed on the CFA (Fig. 9a), enabling stroke induction without disturbing the implanted electrodes. This setup allows direct comparison of pre-and post-stroke oscillation in the same animals. We recorded the local field potential (LFP) in freely moving mice in their home cage 3 weeks after stroke (Fig. 9a). Stroke causes an immediate global spectral power decrease ranging over

delta, theta and gamma frequencies (Supplementary Fig. 16). Recovery of spectral power varied according to the spectral frequency band and the vigilance state. Low gamma power (30–60 Hz) exhibited notable recovery following stroke (Fig. 9b–d), and this was significantly enhanced in animals undergoing rehabilitation (Fig. 9c,d).

Next, we further assessed the association of gamma waves with functional recovery in human stroke patients. We studied 27 patients with stroke admitted to an inpatient rehabilitation facility (IRF) a median of 12 [8–17] days post-stroke (Visit 1, Supplementary Table 1a). Multiple EEGs were recorded during time in the IRF admission, and an additional EEG was obtained approximately 3 months (86 – 103 days) after stroke onset. The EEG was a 3-min recording taken at rest, and relative low gamma power (30–40 Hz) was analyzed in ipsi (i)- and contralesional (c) hemispheres within five motor regions: M1 (iM1 and cM1), dorsal premotor areas (iPMD and cPMD) and midline supplementary motor area (SMA). 17 patients were available for repeat EEG and functional assessment at Visit 5 (median 92 [86 – 103] days). We found that the arm motor Fugl-Meyer (FM) score significantly improved from Visit 1 to Visit 5 (Fig. 9f: p = 0.0004). At Visit 1, the relative low gamma power did not differ between patients and 27 age-matched healthy control subjects within any of 5 motor regions (Supplementary Table 1b). Subsequently, low gamma oscillations significantly increased in iM1, cM1 and SMA from Visit 1 to Visit 5 (Supplementary Table 1c). Notably, the increase of gamma power in ipsilesional M1 was significantly correlated with the FM scores in moderate to severe stroke patients during the recovery period (V3-5, V1 FM score <46, Fig. 9g, h). Thus, gamma oscillations increase in stroke patients during rehabilitation recovery after stroke, as they do in the mouse. Together, these data support the hypothesis that rehabilitation enhances functional recovery through PV interneuron-mediated mechanisms, and an oscillatory signal of increased neuronal activity by the gamma rhythm.

## Activation of PV interneuron improves functional recovery

Finally, we assessed whether there is a pharmacological approach to simulate rehabilitation-induced recovery from stroke, through activation of PV interneuron circuits. To this end, we tested two compounds: AUT00201, a selective positive modulator of Kv3.1 ion channels predominantly found on PV interneurons[60] and DDL-920, a selective negative modulator of the γ-aminobutyric acid type A receptors with α1β2δ subunits (GABAARδ) responsible for the tonic inhibition of PV interneurons[61]. The α1β2δ GABAAR is uniquely expressed by PV interneurons as opposed to the α4β2/3δ GABAAR of dentate gyrus granule cells, thalamo-cortical, cortical/hippocampal pyramidal and medium spiny neurons, and the α6β2/3δ GABAAR predominantly expressed by cerebellar granule cells[62]. Positive modulation of Kv3.1 causes faster activating kinetics and increased firing frequency in fast-spiking GABAergic interneurons[63]. On the other hand, negative modulation of GABAARδ reduces tonic inhibition and enhances the excitability of PV interneurons[64]. These alterations in activity subsequently modulate gamma oscillations[64,65]. We orally administered the drugs, to enhance the applicability of the formulation for potential clinical translation (Fig. 10a). We confirmed that Kv3.1 ion channels and GABAARδ expressed mainly in PV interneurons in both intact and

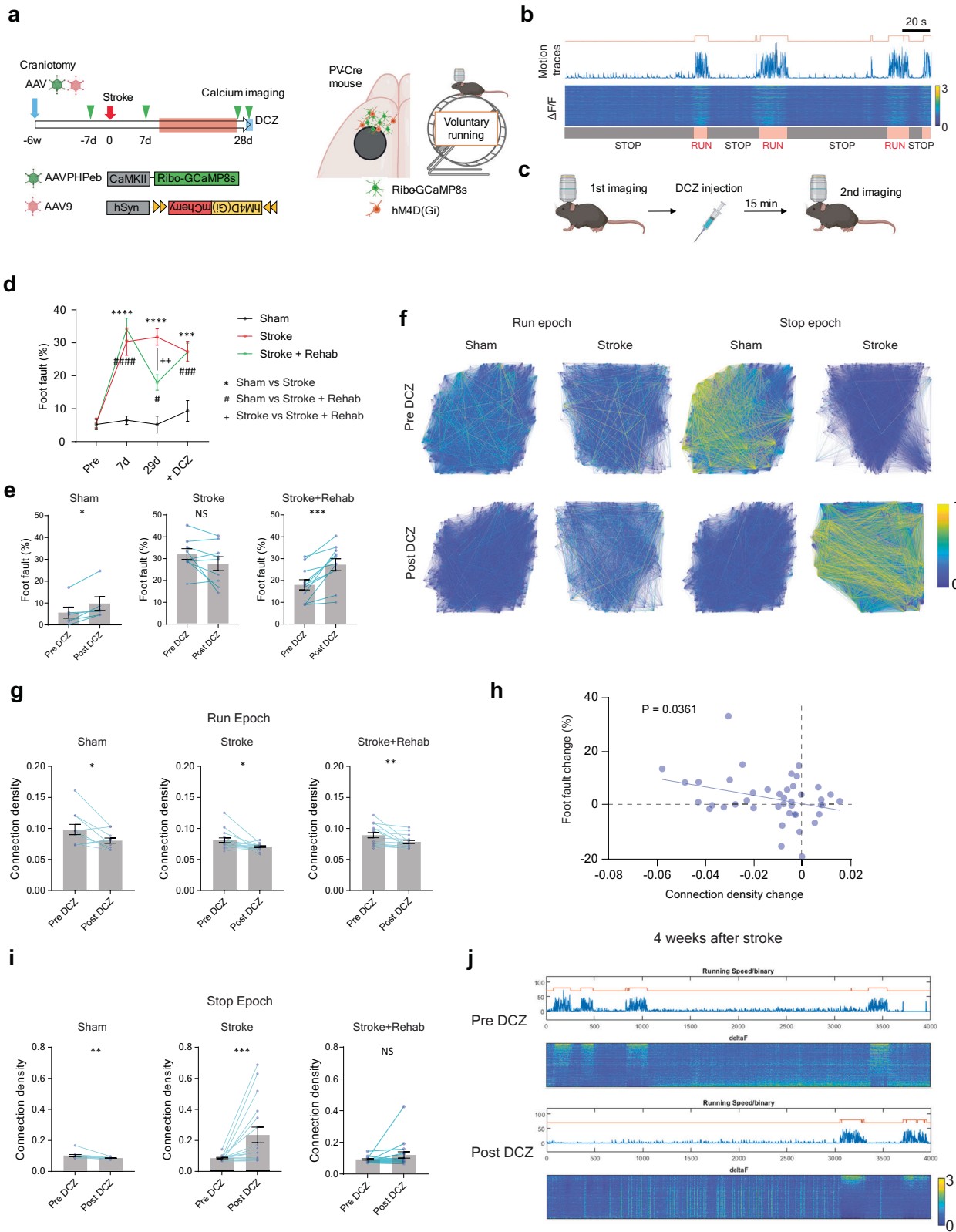

**b**

Motion traces

ΔF/F

STOP   RUN   STOP   RUN   STOP   RUN   STOP

**c**

1st imaging → DCZ injection → 15 min → 2nd imaging

**d**

Foot fault (%)

Pre   7d   29d   + DCZ

- Sham
- Stroke
- Stroke + Rehab

\* Sham vs Stroke
\# Sham vs Stroke + Rehab
\+ Stroke vs Stroke + Rehab

**e**

Sham    Stroke    Stroke+Rehab

Foot fault (%)

Pre DCZ   Post DCZ

**f**

Run epoch                Stop epoch

Sham   Stroke   Sham   Stroke

Pre DCZ

Post DCZ

**g**

Run Epoch

Sham    Stroke    Stroke+Rehab

Connection density

Pre DCZ   Post DCZ

**h**

Foot fault change (%)

P = 0.0361

Connection density change

**i**

Stop Epoch

Sham    Stroke    Stroke+Rehab

Connection density

Pre DCZ   Post DCZ

**j**

4 weeks after stroke

Running Speed/binary

deltaF

Pre DCZ

Running Speed/binary

deltaF

Post DCZ

stroke animals (Supplementary Fig. 17). Activation of PV interneurons was tested by single dosing. As expected from the PV interneuron-selective effects of these drugs, both AUT00201 (20 mg/kg[66], Supplementary Fig. 18) and DDL-920 (10 mg/kg[64], Supplementary Fig. 19) increased the expression of the immediate early gene Zif268 in PV interneurons without changing the density of cells expressing Zif268. Only DDL-920 showed a statistically significant increase (Fig. 10b). In a

stroke-recovery study, we started the drug treatment 3 days after the stroke and evaluated the recovery of forelimb motor function with skilled reach (pasta matrix) and gait (grid walk) tests. We did not observe any adverse effects such as weight loss or motor deficits in either sham or stroke animals. Stroke animals treated with the vehicle and AUT00201 exhibited prolonged disability in precisely retrieving pasta pieces (Fig. 10c, d). In contrast, DDL-920 treatment led to a

**Fig. 8 | PV interneurons regulate functional connectivity in healthy and stroke animals. a** Timeline, virus vectors, and imaging field for calcium imaging with chemogenetic inhibition. **b** Representative motion trace and color-mapped calcium transient. The orange line above the motion trace indicates binary detection of the running epoch. **c** Protocol for calcium imaging and DCZ injection. **d** Foot fault rate during the calcium imaging. Two-way repeated measure ANOVA, time by group, F (6, 69) = 11.04, *P* < 0.0001Sidak's multiple comparison test. **e** Foot fault rate change by DCZ injection. Two-tailed Wilcoxon test (**d**, **e**) n = 6 (Sham), 9 (Stroke) or 11 (Stroke + Rehab). **f** Examples of connection map before and after DCZ

injection in Sham and Stroke groups. **g** Connection density changes by DCZ injection in the run epoch. Two-tailed Wilcoxon test. *n* = 11 (Sham), 16 (Stroke) or 15 (Stroke + Rehab). **h** Correlation between the motor performance change in the grid walking and the connection density change after the DCZ injection. Pearson correlation, two-sided (n = 42, r = 0.324, *P* = 0.0361). **i** Connection density changes by DCZ injection in the stop epoch. Two-tailed Wilcoxon test. n = 12 (Sham), 17 (Stroke) or 18 (Stroke + Rehab). **j** Representative recordings before and after the DCZ injection in stroke animals. c *P < 0.05, **P < 0.01, ***P < 0.001. Data are presented as means ± sem. (**a**, **b**) Created in BioRender[127].

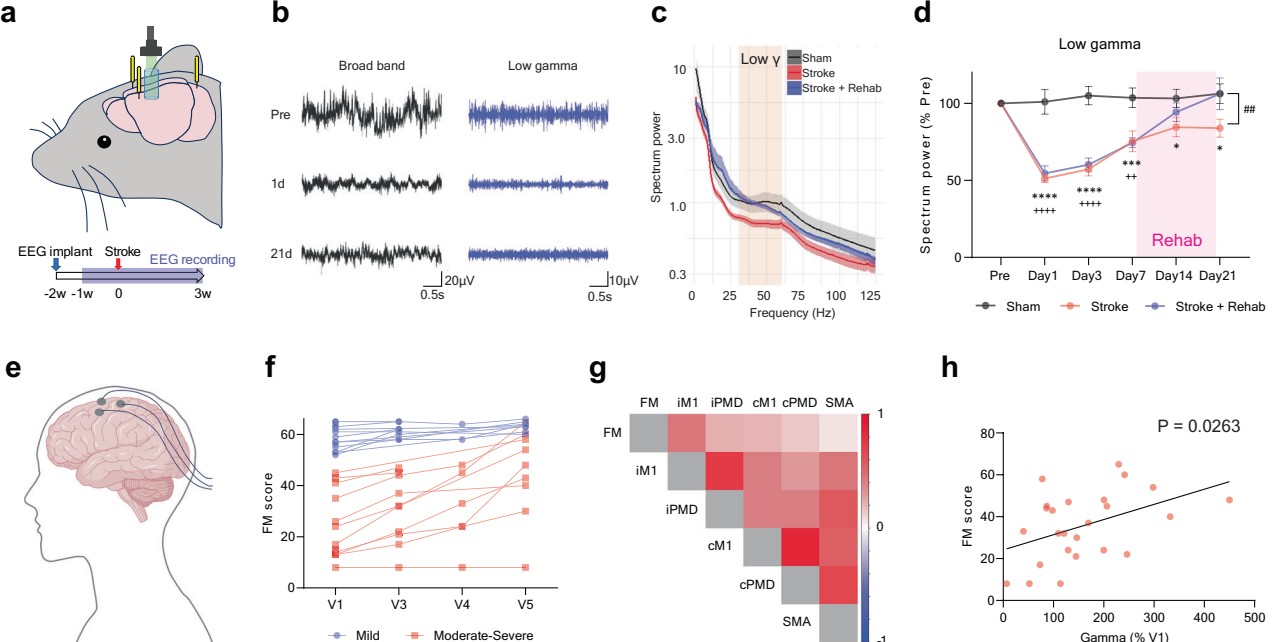

**Fig. 9 | Gamma oscillation in the stroke mouse model and human stroke patients. a** Electrode placement and timeline for the EEG recording. **b** Representative LFP traces in stroke animals. **c** Normalized power spectra of network oscillation in ipsilesional premotor cortex in awake period mice 21 days after the stroke (means ± sem). The orange rectangle indicates the lower gamma frequency band. **d** Normalized spectrum power change in low gamma frequency. Mixed-effects model, time by group, F (10, 62) = 11.87, *P* < 0.0001. *P < 0.05, ***P < 0.001, ****P < 0.0001: Sham vs Stroke, ++ P < 0.01, ++++ P < 0.0001: Sham vs Stroke + Rehab, ## P < 0.01: Stroke vs Stroke + Rehab, Tukey's multiple

comparisons test. Two-sided. Tukey's HSD correction for multiple comparison. n = 4 (Sham), 7 (Stroke) or 5 (Stroke+Rehab). **e** Illustration of human EEG. **f** Arm motor Fugl-Meyer (FM) score after stroke. **g** Correlation matrix of the FM score and the relative gamma power during the recovery period (V3-5) in the ipsi (i) and contralesional (**c**) motor related areas. M1: primary motor area, PMD: dorsal premotor area, SMA: supplementary motor area in moderate to severe stroke patients (V1 FM < 46). **h** Correlation between the FM score and the normalized gamma power. Spearman correlation, two-sided (n = 25, r = 0.443, *P* = 0.026). **e** Created in BioRender[127].

complete recovery of motor function after stroke (Fig. 10c, d). AUT00201 and DDL-920 treatment also produced faster recovery in the grid walk test (Supplementary Fig. 20). These data establish the principle that pharmacological agents can drive beneficial cellular effects seen in rehabilitation-induced stroke recovery and promote behavioral recovery equivalent to that seen in rehabilitation-induced stroke recovery.

## Discussion

This study found that motor systems in the brain have a rehabilitation-induced cellular circuit that mediates stroke recovery by selectively increasing synaptic connections between stroke-projecting neurons and PV interneurons in the premotor cortex. Rehabilitation-induced functional recovery relies on neuronal activation in these neuronal populations. Post-stroke motor recovery is associated with increased neuronal connectivity and gamma oscillations, in both mouse models and humans. Both neuronal connectivity and gamma oscillations are regulated by PV interneurons. Collectively, our results indicate that synaptic connections between stroke-projecting neurons and PV interneurons contribute to functional recovery by restoring neuronal

synchronization in the ipsilesional premotor cortex. In addition to the regulation of neuronal activity patterns, PV interneurons may also be involved in neuronal plasticity to reorganize neuronal circuits. PV interneurons behave in a manner analogous to their regulation of a developmental critical period in the postnatal brain, with similar alterations in peri-neuronal nets after rehabilitation[67]. This cellular platform for post-stroke rehabilitation led to the identification of a drug that reproduces the beneficial effects of rehabilitation on behavioral recovery after stroke. A rehabilitation drug could provide substantial benefits in clinical stroke recovery[68].

Stroke induces a series of plasticity events at the network, circuit, cellular, and molecular levels. At the molecular level, stroke induces upregulation of growth-promoting gene expressions in parallel with an excitation/inhibition imbalance alteration[69]. These molecular changes lead to functional and morphological changes in cellular components such as dendrites, axons, and synapses to reconnect the dissociated circuits. These plasticity events are the most prominent in the peri-infarct tissues, but the brain areas anatomically connected to the stroke site also engage in restorative plasticity, especially with therapeutic interventions. For example, corticospinal neurons in the

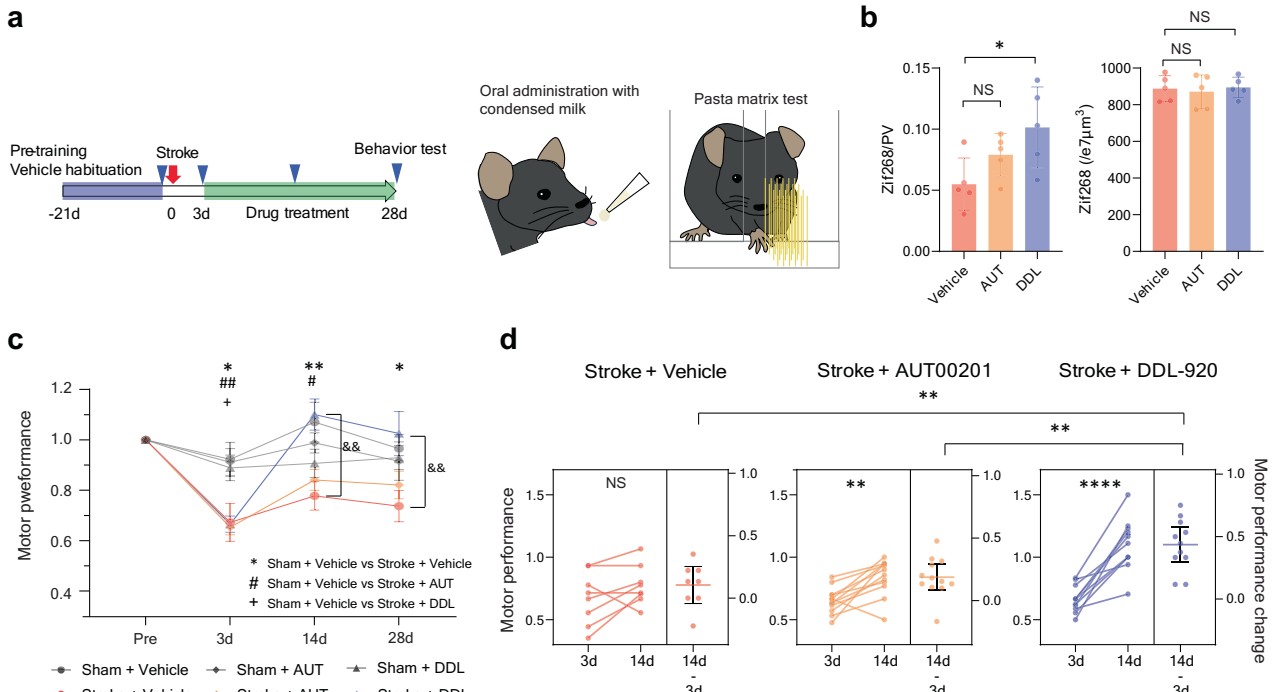

**Fig. 10 | PVIN targeting drug therapy reproduces rehabilitation effects.**
**a** Timeline and procedure for the drug administration of AUT00201 ('AUT', 20 mg/kg p.o.) or DLL-920 ('DDL', 10 mg/kg p.o.) and behavioral test. **b** Ratio of Zif268 positive PV interneurons (Left, F (2, 12) = 4.332, P = 0.0383) and density of Zif268 positive cells (right, F (2, 12) = 0.1324, P = 0.8773). One-way ANOVA, Tukey's multiple comparisons test. n = 5. **c** Normalized motor performance in the pasta matrix test. Two-way repeated measure ANOVA, time by group, F (15, 189) = 3.890, P < 0.0001. *P < 0.05, **P < 0.01: Sham + vehicle vs Stroke + Vehicle, #P < 0.05,

##P < 0.01: Sham + vehicle vs Stroke + AUT, ++P < 0.01: Sham + vehicle vs Stroke + DDL, & &P < 0.01: Stroke + vehicle vs Stroke + DDL, Sidak's multiple comparison test. **d** Functional recovery from day 3 to day 14. Two-tailed paired t-test for comparisons of 3 d and 14 d. One-way ANOVA and Tukey's multiple comparison for comparisons of functional gains. F (2, 28) = 8.967, P = 0.0010. **P < 0.01, ****P < 0.0001. **c, d** n = 12 (Sham+Vehicle), 12 (Sham+AUT), 14 (Sham+DDL), 8 (Stroke+Vehicle), 12 (Stroke+AUT), 11 (Stroke+DDL). Data are presented as means ± sem.

contralesional hemisphere contribute to functional recovery through axonal sprouting from intact to denervated spinal hemi-cord with anti-Nogo-A antibody therapy or optogenetic stimulation[33,70]. Optogenetic and chemogenetic stimulation targeting a specific neuron type or projection, such as VIP interneurons[24] and thalamocortical projection neurons[71], can also enhance functional recovery associated with functional and morphological changes in the stimulated neuronal population. These studies raise the concept that restoring weakened connections or activity (e.g., corticospinal axons, thalamocortical axon, VIP responsiveness) promotes functional recovery.

In the current study, we demonstrated that stroke-projecting neurons are involved in functional recovery after a stroke, seemingly through the restoration of synaptic inputs. We observed that rehabilitation significantly increased synaptic inputs from the contralesional hemisphere and also attenuated synaptic input loss from some brain areas, including the peri-infarct CFA and thalamus. Since the reaching task requires neuronal activity in the ipsi- and contralesional motor cortex[72] and thalamus[26,73], rehabilitation-induced repetitive neuronal activity may restore neuronal connectivity via activity-dependent plasticity. Furthermore, we found that rehabilitation increases synapses formed by PV interneurons and the stroke-projecting neurons, whereas stroke did not significantly reduce the number of the PV synapses or PV interneurons in the RFA. Notably, stroke disturbed the fractional composition of PV interneuron synapses over total GABAergic synapses on the stroke-projecting neurons, and rehabilitation adjusted the fractional disturbance. Additional patch-clamp experiments further validated this finding from a functional perspective. Subtype-selective inhibitory synaptic plasticity occurs in some contexts, such as exposure to a novel environment and motor learning[30,31]. Furthermore, subtype-selective inhibitory neuron

abnormalities have been observed in pathological conditions, including Alzheimer's disease[74] and schizophrenia[75]. Collectively, these findings suggest that inter-interneuron balance (composition of inhibitory synapse across interneuron subtypes) may impact functional recovery after stroke in addition to excitation/inhibition balance.

As the stroke-projecting neurons lose their synaptic output partner (CFA), forming new synaptic connections with remaining axonal projections or by axonal sprouting might be a reparative response. Stroke-projecting neurons project their axons to the contralateral cortex, striatum, thalamus, cerebral peduncle, and peri-infarct cortex. Our previous research revealed that transcriptional modulation of the memory/learning system, such as CREB overexpression[76] and CCR5 knockdown[77], increases axonal sprouting to the peri-infarct motor cortex and the contralesional hemisphere. Rehabilitation is a relearning process of previously learned behaviors that may promote similar axonal projections. Anterograde axonal tracing combined with synaptic labeling (e.g., GRASP) and optogenetic validation is warranted for future study.

In addition to structural plasticity after stroke, the current study underscores the notable complexity of synaptic input dynamics. While EPSC frequency analysis showed a marked increase in animals after rehabilitation—aligning with dendritic spine data—we also observed a notable increase in EPSC amplitude, specifically in stroke animals. Moreover, IPSC frequency rose post-stroke, even though the total number of VGAT puncta remained unchanged. Synaptic input frequency and amplitude are influenced not solely by synapse number but also by the functional state of each synapse, shaped by factors such as intrinsic neuronal excitability, presynaptic release probability and postsynaptic modifications, including enhanced receptor sensitivity and upregulation of postsynaptic receptors. Previous stroke research

demonstrated that brain ischemia induces AMPA and NMDA receptor clustering[78]. Stroke also induces robust transcriptional changes, including synaptic protein and neurotransmitter receptors[79]. Furthermore, stroke-projecting neurons lose their axons at the stroke site. This generates retrograde injury signals that, in turn, initiate extensive transcriptional changes. Prior research has demonstrated that these retrograde injury signals increase vesicle release rates at excitatory synapses[46]. Given the extensive gene expression changes induced by stroke and retrograde injury signals, complex transcriptomic mechanisms are likely to contribute to the observed alterations in excitatory and inhibitory inputs. Deciphering these mechanisms would benefit from comprehensive and systematic approaches like transcriptomic and proteomic analyses combined with whole-cell/cell-attached patch-clamp recordings, and optical methods for establishing unbiased measurements of membrane voltage.

At the circuit level, our calcium imaging study reveals dynamic shifts in neuronal activity and functional connectivity. Seven days post-stroke, there is a marked reduction in the number of active neurons, activation frequency, connection number, and connection density. During the recovery phase, active neurons and connection numbers spontaneously recover, yet the lowered frequency and connection density remain. Rehabilitation enhances the number of active neurons and connections. Fractional connection density analysis shows that rehabilitation reduces the proportion of neurons with sparse functional connectivity, suggesting that rehabilitation helps restore severely affected neurons. These findings imply that rehabilitation selectively modulates the function of initially inactive and severely affected neurons.

Notably, rehabilitation induced overshoot effects, surpassing physiological levels observed in Sham animals across various experiments (e.g., PV synapse number in histology and active neuron count in calcium imaging). During post-stroke recovery, numerous biological metrics exceed baseline physiological levels in brain tissue, particularly with enhanced plasticity, including axonal projections[77], dendritic spine length and turnover[80,81], and blood vessel density[82]. These phenomena reflect two key aspects of stroke biology. First, certain histological and physiological features may indicate enhanced plasticity beyond normal levels, such as increased expression of immediate early genes and decreased expression of perineuronal nets. Since PV interneurons are crucial for synaptic plasticity, their overshoot could signify a transient boost in plasticity, as observed in motor learning[30]. Second, functional recovery often demands more than the original physiological capacity, as the remaining components of the network must compensate for lost functions. This necessity could explain the overshoot in active neurons and connection numbers observed in our calcium imaging study.

Our EEG data revealed dynamic spectral power changes across broad spectral ranges and different vigilance states. We found that spectral power spontaneously recovers to some degree in most spectral ranges. These changes are probably associated with recovery from early depression of neuronal activity or diaschisis. Additionally, previous studies have identified several factors linked to the reduction of gamma oscillations following stroke, including increased tonic inhibition[83], glial activation[83,84], and reduced blood flow[85]. These mechanisms may, therefore, also play a role in the observed effects. Spectral power recovery is incomplete in many state/spectral range combinations. In recordings 21 days after stroke, the stroke animals show significantly lower delta power in all vigilant states. Theta power was also persistently reduced in the NREM/REM, while reduction of low gamma was prominent in the AWAKE state. In the physiological state, theta oscillation is strong in the REM and NREM periods, and low gamma is strong in the AWAKE period. Thus, the reduction of spectral power may persist in the vigilant state in which strong oscillatory activity emerges. Our rehabilitation paradigm induced statistically significant improvement only in the low gamma in the AWAKE period. Accumulating data support the idea that PV interneurons mediate the recovery of low gamma during rehabilitation. Firstly, PV interneurons are considered the major players in generating or regulating the temporal structure of neuronal oscillation, especially gamma oscillation[58]. PV interneurons are adapted for fast synchronization of network activity, as they resonate at gamma frequencies and exert strong perisomatic inhibition that is capable of precisely controlling spike timing[86]. Optogenetic studies further evidence the critical role of PV interneurons in gamma oscillation[87,88]. Moreover, a recent study demonstrated that gamma oscillation plasticity (gamma potentiation) is also mediated by parvalbumin interneurons[89]. Secondly, motor execution activates gamma oscillation, known as motor gamma oscillation[90]. The current study demonstrates that rehabilitation induced immediate early gene expression in PV interneurons is concomitant with enhanced gamma oscillation. An in vivo opto-tagging experiment also revealed that skilled reaching evokes PV interneuron activities[91]. Mouse stroke studies demonstrated that neuronal activation at gamma frequency by light flicker[92] (administered 2 h after the stroke, 1 h, twice daily for 14 days) and optogenetic stimulation targeting vGAT-positive interneurons[85] (administered 3 to 5 min after the stroke, total 48 min) induced persistent enhancement of gamma oscillation. Although these interventions were early after the stroke, these results suggest that the activation of neuronal circuits at gamma rhythm can induce persistent gamma oscillation enhancement after stroke. Lastly, some rehabilitation-induced PV interneuron changes can impact gamma oscillation. We found that rehabilitation increased synapse formation between the stroke-projecting neurons and the PV interneurons and decreased perineuronal net expression. Lensjo et al. reported that the degradation of perineuronal nets by ChABC increases gamma oscillation concomitant with heightened ocular dominance plasticity[93]. Although we found PV synapse formation in specific neuronal circuits, the perineuronal net change might impact the plasticity of the entire cortical network. Collectively, it is probable that PV interneurons are involved in the increased gamma oscillation during rehabilitation. We note that our study specifically focuses on the neuronal circuits formed by stroke-projecting neurons and PV interneurons. Therefore, it is highly plausible that other neuronal networks, including circuits involving different interneuron types, also contribute to the observed increase in gamma power during rehabilitation. To fully understand the mechanism by which stroke and rehabilitation alter neuronal oscillations, mechanistic studies such as chemogenetic and optogenetic interventions, together with EEG recordings should be done to systematically and precisely describe the causal links.

In the current study, we investigated PV interneuron function using acute and chronic DREADD inhibition. We should note that different functionalities are affected by the different DREADD protocols. Acute inhibition of PV interneurons directly affects neuronal signal dynamics in the circuit in which PV interneurons regulate functional connectivity. Conversely, chronic inhibition of PV interneurons would influence plasticity changes in the circuit repeatedly activated by rehabilitation. Activity-dependent mechanisms regulate inhibitory synapse plasticity[94,95]. A recent study also demonstrated that chronic inhibition of PV interneurons abolishes PV bouton formation induced by environmental enrichment in a model of schizophrenia[96]. Therefore, chronic PV inhibition during rehabilitation is likely to inhibit synapse formation between PV interneuron and stroke-projecting neurons and affect functional connectivity in the circuit.

Two candidate drugs induced distinctive effects depending on the behavior tested. Complete recovery occurred in the pasta matrix test only in the DDL-920-treated animals but not the AUT00201-treated animals. Because immediate early gene expression was significantly upregulated only by DDL-920, improvement of this task may be attributed to the extent of PV interneuron activation. Improvement by DDL-920 was more dramatic in the pasta matrix test, which requires precise reach-to-grasp movement as in our rehabilitation paradigm, compared to the grid walk test. These findings suggest that motor system PV interneuron circuits are responsive to the pattern of motor activity to which they are presented, with reach-to-grasp behavior

plasticity greater than that in walking precision. Further studies are warranted to refine dosing and treatment regimen and duration because the effective level of target occupancy achieved during the study may have been sub-optimal for one or both agents based on the once-per-day dosing strategy employed. The potential for these and other targeted pharmacological agents to work in an additive or synergistic fashion to enhance behavioral outcomes will also be important to evaluate. Additionally, further investigation is necessary to identify how these drugs improve functional recovery, including Kv3.1-mediated PV interneuron firing rate changes, GABAARδ-mediated tonic inhibition, and gamma oscillation.

The current study has some limitations regarding the selection of animals used. We conducted the study with young male mice with motor deficits in the skilled reaching behavior. This approach allows rapid discovery experiments of circuit and drug effects. About 80 % of strokes occur in individuals over 65[97]. Aging is associated with increased stroke prevalence, greater stroke-related mortality, and disability, and older patients are at higher risk of complications related to thrombolytic treatment compared to younger patients[97]. Aging also affects stroke-related biology, including neuronal cell death[98], synaptic plasticity[99], and immunological response[100,101]. Sex differences can also influence stroke outcomes. Several studies reported worse disability and poorer quality of life in female stroke patients[102,103], possibly due to different cerebrovascular[104] and immune responses[105]. Therefore, confirming the significance of neuronal circuits formed by stroke-projecting neurons and PV interneurons in different ages and genders is crucial for translating the current findings into clinical settings. We also selected animals with detectable motor disability in the behavior tests because our goal is the development of therapies for severely affected stroke patients who have little chance for satisfactory recovery. Previous studies have demonstrated that recovery from motor impairment is not necessarily correlated to the size of the brain lesion but is better correlated to cortical remapping and axonal sprouting[106,107]. We identified functional connectivity in the premotor cortex as a strong predictor of functional outcomes. Furthermore, the human EEG study reveals a significant correlation between gamma power increase and motor recovery in patients with moderate to severe motor disability. These findings suggest that local neuronal ensembles in the remaining network might be responsible for motor impairment in severely affected individuals.

Our findings have clinical implications. First, stroke or rehabilitation-induced network changes are heterogeneous, even within a single brain region. The comparisons of corticospinal neurons and stroke-projecting neurons and three main interneuron subtypes support this concept. Identification of precise neuronal circuit mechanisms and cell type-specific drugs may allow selective connectivity modification that activates pro-recovery circuits and inhibits anti-recovery circuits. Second, neuronal synchronization in circuits adjacent to or connected with stroke is associated with functional recovery. Similar strategies have been suggested in other neurological diseases[108]. These data suggest that establishing an envelope of increased neuronal excitability, with the restoration of gamma oscillations, may facilitate task-specific motor training in promoting neuronal and behavioral recovery after stroke. Finally, our study underscores the potential of drug therapies that replicate the biological processes underlying rehabilitation mechanisms for functional recovery. While rehabilitation is a modestly effective therapy, various obstacles hinder its effectiveness or patient participation. Our rehabilitation paradigm models human rehabilitation. Therefore, the effects of rehabilitation in the study are not as robust as other unphysiological modalities, such as gene knock-out or optogenetics, capturing physiological rehabilitation characteristics. These characteristics could be challenging when detecting significant biological events but enable the identification of clinically relevant biological changes in physiological conditions. A deeper understanding of rehabilitation biology holds the promise of developing a medical rehabilitation drug with a more effective treatment for stroke patients experiencing motor disabilities, especially for those unable to engage in quality rehabilitation therapy.

## Methods

### Mice

All procedures were performed under an NIH-approved animal protocol and the University of California Los Angeles Chancellor's Animal Research Committee (ARC-2000-159-AM-011). 2–4 month-old adult C57BL/6 (The Jackson Laboratory) or B6 PV-Cre (B6;129P2-Pvalbtm1(cre) Arbr/J, The Jackson Laboratory) male mice were maintained on a 12 h light/dark cycle with free access to food and water except for the periods during behavior tests and rehabilitation for chronic DREADDs experiments. Total 649 mice (C57BL/6 = 569 and B6 PV-Cre =80) were used in this study (Supplementary Table 3). All procedures, including stroke surgery, behavioral tests, histological analysis, 2-photon calcium imaging analysis, and EEG analysis were conducted by the blinded experimenters, except for rehabilitation treatment. We determined the sample size based on the previous studies in our lab that assessed the effects on behavioral recovery after a stroke.

### Recombinant AAV cloning and production

Adeno-associated viruses serotype PHP.eB were packaged using the previously published protocol[109]. All the plasmids used for the backbone and the insert were purchased from Addgene. The backbone and the insert sequences were assembled by Gibson assembly. AAV PHP.eB were purified from HEK293T cells transfected with pAAV (Supplementary Table 2), capsid, and helper plasmids using polyethyleneimine transfection. The media was changed the following day, then media was collected at 3 and 5 days post-transfection, and cells were collected at 5 days. Cells were pelleted and resuspended in 500 mM NaCl, 40 mM Tris, and 10 mM MgCl$_2$, supplemented with salt-activated nuclease (100 U/mL, ArcticZymes), and incubated for 1 h at 37 °C. Viral particles were precipitated from media via incubation with 8% PEG8000 at 4 °C for 2 h followed by 30 min centrifugation at 4000 × g for 30 min at 4 °C; the resulting pellet was also resuspended in salt-activated nuclease buffer and incubated with the cell lysate for 30 min at 37 °C. Iodixanol density gradients were poured from the top, 15%, 25%, 40%, and 60% steps; 25% and 60% steps included ~1% phenol red for visualization. Lysates were then pelleted at 3000 × g for 15 min and loaded onto iodixanol gradients. Gradients were spun at 350,000 × g on an ultracentrifuge for 2 h 25 min at 18 °C (Beckman Coulter). After the ultracentrifuge, the viral layer was removed using a syringe with an 18 G needle and loaded into the filtration column in combination with 10 mL DPBS + 0.001% Pluronic F-68 through a 0.22-μm syringe filter (Millipore). Amicon filtration columns (Millipore, 100 K MW) were used to exchange the remaining iodixanol from the purified virus. Columns were covered with 15 mL DPBS + 0.01% Pluronic F-68 and centrifuged at 3000 × g for 3 min. The centrifuge was repeated 4 times with buffer exchange. In a final centrifugation, the virus was concentrated to its final volume. The virus was titered using qPCR and stored at −80 °C until use.

### Photothrombotic stroke

We induced cerebral ischemia using the photothrombotic model on the CFA contralateral to the dominant forelimb, as determined by the skilled reaching task. Stroke was induced in the left hemisphere if this behavior test was not performed. Mice were anesthetized with 2% isoflurane anesthesia and placed in a stereotactic apparatus. Rectal temperature was monitored and maintained at 37 °C by a heating pad. After exposing the skull through a midline incision, Rose Bengal (1% in saline, 0.01 mL/g of body weight) was administered intraperitoneally. Five minutes after the injection, the brain was illuminated with green laser (20 mW; CLD1010LP, Thorlab) through the intact skull for 10 min. Laser illumination was made centered at 1.5 mm lateral and 0.25 mm rostral to the bregma for CFA stroke. For the second injury on the RFA, laser illumination was made 1 mm lateral and 2 mm rostral to the

bregma for RFA stroke based on the forelimb representation of the C57BL/6 mouse motor cortex[110].

### Rehabilitation

Post-stroke rehabilitation was carried out with a plexiglass reaching box consisting of a chamber with a central table for the mouse to climb onto and millet seed containers on either side (Supplementary Fig. 1). After the stroke, mice with the same dominant hand for the skilled reaching test were paired in a cage. Previous studies in both human patients and animal models reported that social isolation or single housing negatively affects stroke outcomes, such as increased mortality[111,112] and depression-like behavior[113]. Therefore, we avoided isolated housing conditions. Two reaching boxes per cage were set, and 4 g of millet seeds were filled in the container on the dominant limb side every day, five days a week, for 3 weeks, starting day 10 after stroke. As a control procedure, mice were given the reaching box without millet seed.

### Dendritic spine analysis

Morphological changes of dendritic spines were analyzed in the sparsely labeled stroke-projecting neurons and corticospinal neurons. For sparse labeling of neurons, 50 nl of AAV1-pCAG-FLEX-EGFP (1.0x10E13/mL) was injected in the RFA (left side, 1.0 mm lateral, 2.0 mm rostral to the bregma) and 10 nl of retrograde AAV-pkg-Cre (9.3 x 10E10/mL) was injected either the CFA (left side, 1.5 mm lateral, 0.25 mm rostral to the bregma) or the spinal cord (C6-7, right side, 0.4 mm lateral from midline, depth 0.8 mm) 3–4 days before stroke. Mice were perfused with 4% paraformaldehyde 28 days after stroke, and 100 um sections were cut on a vibratome in the sagittal plane. The sections were then processed with tissue clearing agent SeeDB2 (Wako) as indicated in the manufacturer's protocol and mounted on a slide glass. Images were taken using a confocal microscope (Nikon C2) with a 20x lens (NA = 0.75) for neuron localization (neuron depth and distance from the infarct) and a 100x oil immersion lens (NA = 1.40) for dendritic spine analysis. We analyzed neurons with clear apical dendrites in layer 5 (deeper than 500 μm from the brain surface) or layer 2/3 (150–400 μm from the brain surface) of the RFA. Dendritic spine images were taken with high-resolution 2048 ×512 image stacks (0.2 μm Z-steps) using a 488 nm laser. The images were analyzed with Imaris software (Bitplane). The filament tracer plugin was applied to reconstruct the dendrite and spine structure with spine thickness and spine head diameter thresholds of 0.3 μm. The reconstructed structures were reviewed carefully by the blinded experimenter to the group, and dendritic spines, which are included in the dendritic shaft, were excluded from the analysis.

### Monosynaptic tracing

Transsynaptic tracing studies were carried out as described previously with slight modifications[114]. For rabies virus tracing, 50 nl of retrograde AAV-pkg-Cre (9.3 x 10E12/mL) was injected either CFA (left side, 1.5 mm lateral, 0.25 mm rostral to the bregma) for stroke-projecting neurons or spinal cord (C6-7, right side, 0.4 mm lateral from midline, depth 0.8 mm) for corticospinal neurons 3–4 days before stroke. 200 nl of a 1:1 volume mixture of AAV1-Esyn-DIO-TVA-YFP (1.13 x 10E11/mL) and AAV8-FLEX-H2B-GFP-2A-oG (4.57 x 10E12/mL) was injected into the RFA 10 days after stroke. Then, G-deleted rabies virus (RVdG, 1.28 x 10E8/mL) was injected into the RFA 28 days after the stroke. Mice were perfused 5 days after rabies virus injection with 4% paraformaldehyde. Brains were dissected and post-fixed for 24 h and placed in 30% sucrose for 48 h. They were embedded in tissue compounds (O.C.T Compound, Fisher Scientific) and stored at −30 °C until sectioning. We sectioned 50-μm coronal sections and collected them in a 24-well culture plate containing an anti-freeze solution.

Images were taken in 21 sections with 300 μm intervals from the rostral-most tip of the cortex to 6.3 mm caudal (approximately 3.1 mm to −3.2 mm anterior to the bregma in a brain atlas) using an all-in-one fluorescence microscope (10× objective: BZ-X700; Keyence). mCherry-

positive neurons were manually counted through the entire brain, except near the starter cell location (RFA). mCherry positive input neurons were assigned to specific brain regions based on classifications of the Allen Brain Atlas (http://mouse.brain-map.org/static/atlas), using anatomical landmarks in the sections except motor areas (primary and secondary motor areas). Motor areas were classified into RFA and CFA for consistency with the other experiments. For ipsilesional RFA, where densely mCherry and GFP positive cells exist, images were taken using a confocal microscope, and the number of mCherry-positive cells and mCherry/GFP double-positive cells (Starter cells) was counted using Imaris software (Bitplane).

### Immunohistochemistry

Sections were washed twice with PBS and then incubated with 0.01 M citrate buffer for 30 min at 80 °C for antigen retrieval. After incubation with a blocking buffer containing 2% BSA 0.1% TritonX-100 in 0.01 M PBS for 30 min, the sections were incubated for 72 h at 4 °C with the primary antibody dissolved in the blocking buffer. Next, the sections were washed 3 times with PBS containing TritonX-100 and incubated with Alexa Fluor-conjugated secondary antibodies at 4 °C overnight. The sections were washed 3 times with PBS and mounted on a gelatin-coated slide glass with aqueous mountain medium (Fluoromount™, Sigma).

### In situ hybridization

In situ hybridization was performed according to the vendor's instructions (HCR™ probe: Molecular Instrument) with slight modification. Sections were washed twice with 5xSSC and then incubated with probe hybridization buffer for 30 min at 37 °C. After incubation with a probe hybridization buffer, the sections were incubated for 24 h at 37 °C with the HCR probe dissolved in the blocking buffer. Next, the sections were washed 4 times with probe wash buffer at 37 °C and 3 times with 5xSCC at room temperature. After incubation with an amplification buffer for 30 min, the sections were incubated with a hairpin amplifier for 24 h at room temperature. The sections were washed 5 times with 5xSCC and mounted on a gelatin-coated slide glass with aqueous mountain medium (Fluoromount™, Sigma).

### Quantification of immediate early gene immunostaining

For detection of immediate-early gene expression after rehabilitation, mice received an injection of retrograde AAV-hSyn-EGFP (1.0 x 10E13/mL, 50 nl) one week before the stroke and started rehabilitation training 10 days after the stroke. The mice were perfused 48 h after the initiation of rehabilitation training according to the previous paper which demonstrated maximum immediate early gene expression at 48 h after motor training[115]. We initiate rehabilitation training by providing a rehabilitation box filled with millet seeds at 6 pm, 10 days after the stroke. We refilled millet seeds 24 h later and perfused animals 48 h after the initiation of rehabilitation. Sham-alone and Stroke-alone animals had an empty rehabilitation box as a control treatment. For detection of immediate early genes after AUT00201 and DDL-920 drug treatment, the mice were perfused 2 h after the oral drug administrations. We performed immunostaining using anti-Zif268 or anti-FosB antibody with anti-Parvalbumin antibody. Z-series stack images were taken using a 20x objective in RFA of the 4 sections with 300 μm intervals. Using the particle module in Imaris software, we counted the number of immediate early gene (Zif168 or FosB) positive cells, GFP-positive cells, and parvalbumin-positive cells and calculated the ratio of immediate early gene-positive cells in the stroke-projecting neurons (GFP-positive) and PV interneurons (parvalbumin-positive). All analysis procedures were performed in a blind manner and threshold parameters were kept constant across the groups.

### Quantification of perineuronal nets

For imaging perineuronal net, we perfused the mice after stroke and rehabilitation training for 3 weeks. To stain the perineuronal net

surrounding the PV interneurons, we used biotin-conjugated lectin from Wisteria floribunda (MilliporeSigma) and Cy3-conjugated streptavidin (Jackson Immunoresearch Labs) to visualize the biotin label with anti-parvalbumin immunostaining. We took z-series stack images using a 20x objective in RFA of the 4 sections with 300 μm intervals. Using Imaris software, perineuronal nets were reconstructed by surface rendering and determined the perineuronal net positive PV interneurons with filters; the distance between perineuronal nets and PV interneurons <0 μm.

## GRASP assay

The synaptic connection between stroke-projecting neurons and PV interneurons was labeled with the GFP reconstitution across synaptic partners (GRASP) technique as described previously[116]. Retrograde AAV-pgk-Cre (9.3 x 10E12/mL, 50 nl) was injected in CFA 7 days before the stroke. Mixture of AAV(PHPeB)-S5E2-pre-eGRASP (1.87 x 10E12/ml, 200 nl) and AAV(PHPeB)-EWB-DIO-myrTagRFP-T-P2A-post-eGRASP (2.77 x 10E12/ml, 200 nl) was injected in RFA 21 days after stroke. Our methodological validation confirmed that AAV-S5E2-preGRASP has a PV specificity of over 90%. We also confirmed that a single virus injection of pre-eGRASP or post-eGRASP does not produce any green fluorescent signal, indicating the GRASP signal is specific to the combination of pre and post-GRASP components (Supplementary Fig. 10). The animals were perfused with 4% PFA 28 days after stroke, and 50 μm thick sagittal sections were cut with a vibratome. The sections were then stained with rabbit anti-vGAT antibody and donkey anti-rabbit antibody conjugated with Alexa Fluor 647. Sections were observed with a confocal microscope using a 20x lens to determine the depth from the cortical surface and the distance from the infarct. The sections were then observed using a 100x lens for the detection of the GRASP signal. The images of myrTagRFP, GRASP, and vGAT signals were taken with a Z-interval of 0.5μm and pinhole diameter of 60 μm (8.03 pixels/μm). The soma of the imaged cell was located at the center of the imaging field, which contains the soma and the dendrites with lengths of up to 60 μm. The images were analyzed by Imaris software. The filament tracer plugin and surface rendering were applied to reconstruct dendritic structures of stroke-projecting neurons labeled with myrTagRF. GRASP puncta and vGAT puncta were reconstructed by surface rendering with filters; the distance from the myrTagRFA neuron <0.1 μm and the volume > 0.1 μm³. GRASP puncta colocalized with vGAT puncta was detected by filtering GRASP puncta with the distance from vGAT puncta <0 μm.

## Infarct volume measurement

We measured the infarct volume as we reported previously[117]. Serial brain sections with 280 μm intervals were stained with NeuN and GFAP antibodies, and images were taken using a confocal microscope (4x objective; Nikon). The borders of intact dorsal cortical tissue were outlined with 10 brain sections covering the prefrontal cortex, motor cortex, and sensory cortex (1.5 mm to −1.0 mm anterior to the bregma in MouseBrainAtlas). The ventral end of the dorsal cortical tissue was determined at the center level of the ventricle. The healthy tissue volume was calculated by multiplying the area and the section interval. The brain damage was calculated as follows;

$$\text{Brain damage(mm}^3) = \text{Contralesional cortex volume} \\ - \text{Ipsilesional cortex volume} \tag{1}$$

$$\text{Brain damage(\%)} = \\ (1 - \text{Ipsilesional cortex volume}/\text{Contralesional cortex volume})*100 \tag{2}$$

## Behavioral assessment

**Skilled reaching test.** The skilled reaching test was carried out as previously described with slight modifications[43,107]. We use a Plexiglas reaching box with a tall, narrow window (150 mm high × 5 mm wide) and a 30 mm wide shelf positioned 5 mm above the floor. Mice trained on 5 sequential days per week. Diet was mildly restricted to maintain approximately 90% of normal body weight during the training. Single millet seeds (Hulled Millet; Anthony's Goods) were initially placed within tongue distance, then at gradually greater distances until mice could retrieve the pellet on the indentation 10 mm away from the inside wall. This procedure usually enabled mice to retrieve single pellets using only the preferred forelimb in the first week. After shaping the reaching behavior, mice received 25 trials of training per day for 2 weeks, and pre-stroke motor performance was calculated by averaging the results of the last 2 days prior to the stroke. We examined motor performance in a testing session of 25 trials with a maximum time of 5 min. Mice were allowed to reach up to 3 reaches in each trial. The motor performance score was calculated in the following way:

$$\text{success rate(\%)} = \\ (\text{number of trials in which the pellet was retrieved}/\text{number of trials}) \times 100 \tag{3}$$

Testing sessions were performed as 25 trials with a maximum time of 5 min. We used only animals with a 30% or higher success rate at baseline for the study. Post-stroke motor performance was evaluated for 1 and 4 weeks after stroke. The animals with motor deficits of less than 10% related to the baseline value were omitted from the study.

**Grid walk test.** Mice were tested for the ability to walk on a grid following a previously described procedure[118]. The grid walking device consisted of an elevated metal grid (height 32 cm, length 20 cm, width 26 cm) with a square 13.5 × 13.5 mm mesh (diameter of rungs 1 mm). A mirror was fixed at a 60° angle below the grid, allowing recording from the side and below. Mice were placed onto the grid and were allowed to move for 2 min freely. Mice were trained 1 day before pre-stroke testing and tested 1 day before the stroke and 7 and 28 days after the stroke. Testing trials were video recorded and analyzed frame by frame. The number of foot faults of the contralesional (affected) forelimb in the first 50 steps was counted, and the probability of foot fault was calculated.

**Pasta matrix test.** We conducted the pasta matrix test to assess learned motor performance[119]. All mice were trained for 4 weeks (5 days/week) prior to photothrombotic stroke administration. Diet was mildly restricted to maintain approximately 90% of normal body weight. For the training, mice were placed in the plexiglass chamber with a narrow open slit window (14 ×1/2 inch), and 3.2 cm long pasta pieces were placed in a 5×5 orientation (pasta matrix). During the first week of training, mice were trained for 30 min/day with the pasta matrix set in front of the slit and tilted toward the chamber. During the second week, mice were trained in an upright pasta matrix for 30 min/day. During weeks 3 and 4, the training was conducted for 15 min/day, gradually moving the position of the pasta matrix to the left side of the mouse so that the entire pasta matrix was eventually positioned behind the chamber wall. In this way, most mice can reach the pasta with their right forelimb, which is impaired by the stroke. Motor performance was analyzed by counting the number of pasta pieces successfully retrieved into the chamber as points. Pre-stroke baseline was calculated by averaging the results of the last 2 test sessions prior to the stroke. Motor performance was normalized by the pre-stroke baseline value. We analyzed only the mice that scored at least 7 points on the baseline test and underwent decreased scores after stroke (Baseline > stroke 3 d).

## Chemogenetic experiment

**Acute inhibition.** AAV9-hSyn-DIO-hM4D(Gi)-mCherry (4.6 x 10E12/ml, 200 nl) or AAV9-hSyn-DIO-mCherry (2.1 x 10E12/ml, 200 nl) was injected in the RFA and Retrograde AAV-pgk-Cre (9.3 x 10E12/ml, 50 nl)

was injected in CFA one week before the stroke surgery. The mice received rehabilitation training for 3 weeks from 10 days after the stroke and behavioral tests one day before and 7, 28, and 31 days after the stroke. In the last behavior test, we intraperitoneally injected Clozapine-N-oxide (CNO, 5 mg/kg) 15 min before the behavioral testing.

**Chronic inhibition.** Similar to the acute inhibition study, AAV9-hSyn-DIO-hM4D(Gi)-mCherry (4.6 x 10E12, 200 nl) or AAV9-hSyn-DIO-mCherry (2.1 x 10E12/mL, 200 nl) was injected in the RFA. For stroke-projecting neuron inhibition, retrograde AAV-pgk-Cre (9.3 x 10E12/mL, 50 nl) was injected in CFA in C57BL/6 mice one week before the stroke surgery. For PV interneuron inhibition, B6 PV-Cre mice were used. We intraperitoneally injected Deschloroclozapine (DCZ, 10 μg/kg). Rehabilitation training started 15 min after the injection and continued for 2 h.

**Drug treatment.** AUT00201 (Autifony Therapeutics Ltd; Supplementary Fig. 18) was dissolved in PEG400 at 60 °C, and the resulting solution was added to condensed milk (AUT00201: 10 mg, PEG400: 100 mg, condensed milk: 400 mg, resulting in a final concentration of 2%). DDL-920 (Supplementary Fig. 19) was dissolved in $H_2O$ at 60 °C, and the solution was added to condensed milk (DDL-920: 10 mg, $H_2O$: 600 mg, condensed milk: 400 mg, resulting in a final concentration of 1%). Animals were orally administered the drug-containing condensed milk at a dosage of 1 mg/g body weight (equivalent to 20 mg/kg body weight for AUT00201 and 10 mg/kg body weight for DDL-920). Oral doses of AUT00201 have previously been shown to be active in a mouse model of progressive myoclonic epilepsy on a similar C57/BL6 background[66] and pharmacokinetic studies have demonstrated at 20 mg/kg steady state, after one week of daily dosing, mean peak blood concentrations of approx. 1000 ng/ml are achieved using the formulation employed in this study (corresponding to approx. 0.05uM free in the brain; $T_{max}$ ~ 2 hrs; $t_{1/2}$ ~ 1–2 h; Brain: Blood ratio ~1; Autifony Therapeutics unpublished data on file). These free brain concentrations are sufficient to cause a leftward shift in the Kv3.1 channel activation curve (Supplementary Fig. 18) and would be expected to increase Kv3 activity as a result. Pharmacokinetics of DDL-920 were also evaluated previously[64] and in this study (Supplementary Fig. 19) demonstrating sufficient brain concentration to induce PV interneuron activation. Due to the distinct dissolving requirements dictated by the chemical properties of these drugs, half of the vehicle group received a PEG400-based vehicle, while the other half received an $H_2O$-based vehicle. Ensuring no discernible effects of the vehicle difference in both sham and stroke animals, the data from the two vehicle groups were combined. Oral administrations were consistently performed within the same time window (the first 2 h from the beginning of the dark cycle) every day. Animals underwent habituation to the treatment with the vehicle over 5 consecutive days, commencing 4 weeks before the stroke, and continued to receive treatment once a week thereafter. The animals were treated 5 days a week for 4 weeks, starting 3 days after the stroke.

**Calcium imaging**

**Cranial window implantation.** 100 nl of GCaMP virus (AAV PHPeb-CaMKII-Ribo-jGCaMP8s-WPRE, 6.0 × 10E12/mL) was injected at 2 sites in RFA (left side, 0.8 and 1.3 mm lateral from midline, 2.0 mm rostral to the bregma, depth 0.3 mm). In the chemogenetic experiment, DREADDs or control virus was mixed into the GCaMP virus solution (AAV9-hSyn-DIO-hM4D(Gi)-mCherry, 2.3 × 10E12/mL, AAV9-hSyn-DIO-mCherry, 2.1 × 10E12/mL). A 4 mm craniotomy was made to cover the entire motor cortex, including CFA and RFA. The skull was removed, and a 4 mm circular glass plate (Warner Instrument) and stainless steel headbar (D&A Metal Fabrication) were permanently glued to the skull

using Vetbond (3 M). The skull surface, the glass plate, and the base of the headbar were then covered with dental acrylic.

**Habituation.** We used the walking paradigm for the behavior task in the calcium imaging for three reasons. 1) The previous 2-photon imaging studies demonstrated that the grid walking behavior sensitively detects network dysconnectivity after stroke. Therefore, it is a suitable behavioral paradigm to investigate circuit changes induced by post stroke interventions[28,39]. 2) In our behavior tests, the grid walk test more sensitively detected behavioral differences in rehab-induced recovery than the skilled reaching test. 3) The same behavioral paradigm in both training and behavioral assessment will introduce confounding factors. This is supported by the long history of stroke rehabilitation research, which shows that post-stroke training can cause compensatory movements in the trained behavior that will confound a behavioral outcome measure that is the same as the training or rehabilitation[40–42].

Neuronal activities were investigated either in the forced-walk paradigm or the free-walk paradigm. In both paradigms, the mouse was restrained in the headbar placed under the 2-photon microscopy, while its body was positioned on a 1.25 cm grid wheel with a 21.5 cm diameter. The free-walk wheel could move freely in the anterior-posterior directions, but the forced-walk wheel rounds at 1.46 rounds per minute to force the mouse to walk forward at a constant speed (103 cm/min). The mice were habituated to the head-restrained condition on the grid four weeks after the cranial window surgery. In the forced walk experiment, mice were habituated for 15 min on the free walk grid wheel in the first 2 days and then habituated for 15 min on the forced walk grid wheel in another 3 days. In the free walk experiment, mice were habituated on the free walk grid wheel for 5 days and gradually extended the time (Day1:15 min, Day2:15 min, Day3:20 min, Day4:20 min, Day5:30 min).

**Photothrombotic stroke.** Photothrombotic stroke was applied 6–8 weeks after craniotomy surgery and at least two baseline imaging. We used the same method described above but with weaker laser power (13 mW) because laser exposure is more intense through the cranial window than the intact skull.

**Calcium imaging.** Two-photon laser-scanning microscopy was performed as described previously[28]. A resonant scanning two-photon microscope (Neurolabware) with a Chameleon ultra laser (Coherent) at a fixed wavelength of 920 nm was used. The 8 kHz resonant raster scanner equates to 512 lines at 30 Hz bidirectionally, which results in image capture at 15.49 Hz. Neurons form ensembles in various time windows, spanning tens of milliseconds to seconds depending on the context and brain area[120]. Previous studies have consistently observed reductions in functional connectivity related to motor impairment after a stroke at similar frame rates in the current study[28,121]. These findings suggest that stroke impacts motor-relevant connectivity dynamics at this temporal resolution.

A dichroic mirror (Semrock) is used to filter the captured light and allow for imaging corresponding to GCaMP signals. The objective used was a 16x water-based lens (0.8 numerical aperture; Nikon). Image capture was relayed to Scanbox, an acquisition software that runs through MATLAB (https://scanbox.org/). GCaMP signals were recorded in layer 2/3 (200–250 μm depth) in the RFA. We determined the imaging field (1 imaging field for the forced-walk experiment and 1–3 imaging fields for the free-walk experiment) in the first recording session and continued the recordings with the same imaging fields. The size of the imaging field is 550μm x 1120μm (mediolateral x anteroposterior). Images were acquired for 2000 frames for the forced walk experiment and 4000 frames for the free walk experiment. Animals' locomotion was simultaneously recorded using infrared lights

and infrared-sensitive cameras placed facing the mouse and from the side of the mouse.

**Recording with chemogenetics.** We conducted chemogenetic manipulation of PV interneurons in the free walk experiment. The mice underwent baseline recording followed by percutaneous DCZ injection (100 µg/kg). The post-DCZ recordings were conducted 15 min after the injection, at which the effect of DCZ becomes maximum. In order to repeat the recording precisely at the same position, the mouse remained restrained on the top of the wheel throughout the recordings.

**Data analysis.** Two-photon imaging data were analyzed following our published approach[28] with some modifications. We used the EZcalcium toolbox (Cantu, https://doi.org/10.3389/fncir.2020.00025) for motion correction, region of interest (ROI) detection, and ROI refinement. Cells were classified as active if the cell had at least one calcium event larger than the z-score of 3.5 in the ROI refinement. Calcium transient peaks were detected by applying a MATLAB smoothing and peak detection function to the waveform. Peaks that were not greater than the root-mean-square of the deltaF/F were not considered Calcium transients. Frequency was calculated by dividing the number of identified Calcium transient peaks by the seconds being imaged, resulting in the number of transients per time (Hz). A deconvolution algorithm is applied to all deltaF/F traces to remove the nonphysiological slow decay and to sharpen the calcium transients. The Pearson product-moment correlation coefficient (PCC) for each deconvolved trace is calculated about all other deconvolved paths. Next, a Monte-Carlo simulation is applied that randomly shifts each deconvolved deltaF/F trace. The PCC value for each time-shifted trace is recalculated for the nonshifted deconvolved traces. The random time-shifted simulations are repeated 100 times for each trace. The distribution of all observed PCC values is then generated. The 95th percentile of this distribution is used as a threshold to define a "significant PCC value" for each trace. The original PCC values are compared with this threshold, and only the ROI pairs with values exceeding this threshold are determined as functionally connected, and the number of functional connections was determined for each ROI. Connection density, defined as the ratio of the measured connection number to the total possible connection number, is calculated for each mouse using the following formula.

$$N = \text{Total active neuron number} \qquad (4)$$

$$\text{Connection density} = \text{Connection number}/\{N \times (N-1)/2\} \qquad (5)$$

Similarly, the connection probability of each neuron was also calculated. This is defined as the ratio of the number of connections measured in each neuron to the total number of possible connections in each neuron (which is equal to the number of neurons other than the neuron of interest).

$$\text{Connection probability} = \text{Connection number from the neuron}/N - 1 \qquad (6)$$

In the free walk experiment, motion states (run epochs and stop epochs) were determined using a custom-written Matlab code, which detects traversing bar positions. The speed of bar movement was determined by the differentiation of bar position and used for thresholding the motion state. According to the motion state, we split the calcium imaging data into the run and stop epoch data and each epoch data was analyzed for frame number, neuron number, mean amplitude, mean frequency, functional connection number, and connection density.

**Validation of PV interneuron chemogenetic inhibition.** To validate the efficacy of PV+ interneuron chemogenetic inhibition, we have performed 2-photon imaging of PV-interneuron activity during DREADD inhibition. AAV9-syn-FLEX-jGCaMP8s-WPRE (2.8 × 10E12/ml) and AAV9-hSyn-hM4D(Gi) (2.3 × 10E12/ml) were mixed and 200 nl of mixed virus was injected at 2 sites in RFA (left side, 0.8 and 1.3 mm lateral from midline, 2.0 mm rostral to the bregma, depth 0.3 mm) in PV-Cre mice. Calcium transients were recorded before and 15 min after the DCZ injection. We used the EZcalcium toolbox for motion correction, ROI detection, and ROI refinement. Then, the ROI matching function was applied to extract cells detected in pre and post-DCZ recordings. Since PV interneurons did not show countable spikes, we analyzed deltaF/F changes induced by running movements using the following formula.

Running-induced delta F/F change =
average delta F/F in run epoch/average delta F/F in stop epoch
$$\qquad (7)$$

We analyzed the cells positively tuned to the run epoch in the pre-DCZ recordings and re-analyzed the same cells in the post-DCZ recordings. For the control procedure, we injected saline instead of DCZ and performed the same recording and analysis.

## Patch-clamp recording

**AAV injection.** Retrograde AAV-CAG-tdTomato (2.5 x 10E13/mL, 50 nl) was injected in the CFA (1.5 mm lateral and 0.25 mm rostral to the bregma, 0.8 mm depth) and AAV9-CaMKIIa-hChR2(H134R)-EYFP (3.1 x 10E13/ml, 400 nl) was injected in the thalamus (1.0 mm lateral and 1.46 mm caudal to the bregma, 4.0 mm depth) 2 days before the stroke surgery.

**Slice preparation.** Mice were anesthetized with isoflurane and decapitated following UCLA Chancellor's Animal Research Committee protocol. Coronal 350 µm thick slices were cut on a Leica VT1200S vibratome in ice cold N-Methyl-D-Glutamine (NMDG)-based HEPES-buffered solution, containing (in mM): 135 NMDG, 10 D-glucose, 4 $MgCl_2$, 0.5 $CaCl_2$, 1 KCl, 1.2 $KH_2PO_4$, 20 HEPES, 27 sucrose (bubbled with 100% $O_2$, pH 7.4, 290–300 mOsm/L). Then, slices were incubated at 32 °C in a reduced sodium artificial CSF (ACSF), containing (in mM): NaCl 85, D-glucose 25, sucrose 55, KCl 2.5, $NaH_2PO_4$ 1.25, $CaCl_2$ 0.5, $MgCl_2$ 4, $NaHCO_3$ 26, pH 7.3–7.4 when bubbled with 95% $O_2$, 5% $CO_2$. After 30 min, low sodium ACSF was substituted for normal ACSF at room temperature, containing (in mM): NaCl 126, D-glucose 10, $MgCl_2$ 2, $CaCl_2$ 2, KCl 2.5, $NaH_2PO_4$ 1.25, Na Pyruvate 1.5, L-Glutamine 1, $NaHCO_3$ 26, pH 7.3–7.4 when bubbled with 95% $O_2$, 5% $CO_2$. All salts were purchased from Sigma-Aldrich.

**Patch clamp recordings.** For recording, brain slices were transferred to a submerged recording chamber at 34 °C and perfused at 5 ml/min with normal ACSF (in mM: NaCl 126, D-glucose 10, $MgCl_2$ 2, $CaCl_2$ 2, KCl 2.5, $NaH_2PO_4$ 1.25, Na Pyruvate 1.5, L-Glutamine 1, $NaHCO_3$ 26, pH 7.3–7.4 when bubbled with 95% $O_2$, 5% $CO_2$). Slices were visualized under an IR-DIC upright microscope (Olympus BX-51WI, 20x XLUM-Plan FL N objective), and either whole-cell recordings or cell-attached recordings were obtained from tdTomato positive pyramidal cells by borosilicate patch pipettes (4–6 MΩ, King Precision Glass). For whole-cell recording, the internal solutions (ICS) contain (in mM): 140 Cs-met, 2 $MgCl_2$ 10 HEPES, 0.2 EGTA, 2 $Na_2$-ATP, 0.2 $Na_2$-GTP, pH of the ICS was adjusted to 7.2 with CsOH and its osmolarity was 285–290 mOsm. For cell-attached recording, the pipet solutions contain (in mM): 120 KCl, 11 EGTA, 1 $CaCl_2$, 2 $MgCl_2$, 10 HEPES, adjusted to pH 7.25 with 35 mM KOH (final K+ concentration is 155 mM) and to osmolarity 310 mOsm with 20 glucose. ICS and pipet solutions were stored at −80 °C in 1 ml

aliquots and then the aliquots were thawed to room temperature before each experiment and kept on ice during recording.

Recordings were obtained using an Axopatch 200B amplifier (Molecular Devices, San Jose, CA, USA), low-pass filtered at 5 kHz (Bessel, 8-pole) and digitized at 10 kHz with a National Instruments USB-6221 (National Instruments, Austin, TX, USA). All data were acquired with Strathclyde Electrophysiology software WinWCP (John Dempster, University of Strathclyde, Glasgow, United Kingdom). Data were stored on a hard disk for offline analysis. After achieving stable whole-cell configuration in voltage-clamp, excitatory post synaptic currents (EPSCs) were recorded at a holding potential (Vh) of −70 mV. After EPSC, voltage-gated currents were evoked by applying a depolarizing voltage ramp stimulation from −100 to +300 mV in 1 s for input resistance measurement. Then, we gradually change the Vh from −70 to 0 mV, at the reversal potential of the excitatory glutamatergic currents, to record only the inhibitory post synaptic currents (IPSCs) from the same cell. Optogenetic stimulation was performed during spontaneous IPSC and EPSC recordings, a 5 s light (475 nm) train containing 10 Hz, 10 ms duration light pulses was given to the recording slice through the objective lens. Whole cell capacitance was estimated from fast transients evoked by a 5 mV voltage command step using lag values of 7 μs and then compensated to 70–80%. The series resistance was monitored before and after the recording, recordings with series resistances >20 MΩ or a change >20% during the recording were excluded.

**Event EPSCs/IPSCs detection and measurement.** EPSC/IPSC events were detected in the 30 s recording segments before the optogenetic stimulation. Event frequency, averaged amplitude, 10–90% rise times and 63% decay times of detected EPSCs/IPSCs were measured. Detection and analysis were performed using EVAN (custom-designed LabView-based software).

**Optogenetic evoked EPSC measurement.** Evoked EPSCs by optogenetic stimulation were detected using a custom-written Matlab code. The code extracts 50 timeframes ranging from 0.01 s before the opto-stimulation (−0.01 sec) to 0.09 s after the opto-stimulation (0.09 s) for each stimulation, then calculates the baseline average and standard deviation from −0.01 to 0 s period. The peak value was obtained as the maximum current difference from the baseline average in the period from 0–0.03 s, and evoked EPSC was counted if the peak value was four times larger than the baseline standard deviation.

**Mouse electroencephalogram**
All procedures regarding the mouse EEG were described previously[122,123]. We utilized EEG recordings in the home cage due to several advantages it offers over shorter recordings conducted during learning and active exploration. Home cage recordings allow for prolonged EEG monitoring during unbiased spontaneous behavior, thereby capturing oscillatory activity in a more naturalistic setting. Mice exhibit regular daily activities and periodic increases in gamma power within their familiar home cage environment. Additionally, home cage imaging facilitates recording sleep cycles, including REM and NREM periods. Previous studies have underscored the significance of oscillatory neuronal activity during the sleep cycle in stroke recovery[124,125].

**Electrode fabrication for in vivo recordings.** Electrodes (Plastics One) were stainless steel, with polyimide electrode insulation, ending in a socket that fitted into the custom-made recording system. Electrode length was 2.5 mm and diameters were 125 μm bare and 150 μm insulated. The electrodes were custom-made as follows: a socket (gold-plated stainless steel Amphenol contact, O.D: 1.07 mm, ID: 0.686 mm and length: 7.87 mm) was attached to a flexible 10 mm long PFA Teflon insulated wire. The free tip of this wire was soldered to another 3 mm long Amphenol socket attached to the 2.5 mm electrode. The openings of the sockets

were trimmed to 1 mm length, cleaned from debris, smoothened on the edges and tightened. The three electrodes were assembled into one unit by connecting their sockets with J-B Weld™ two-part epoxy.

**Stereotaxic surgery.** Mice were anesthetized with isoflurane and placed on the stereotaxic apparatus. The scalp was removed and the two recording electrodes were implanted in the RFA (1.2 mm lateral from midline, 2.0 mm rostral to the bregma, depth 0.3 mm). The electrical reference/ground electrode was implanted over the cerebellum. In addition, a transparent acrylic column with a diameter of 3 mm was embedded as a laser light path for stroke induction. The skull, the lower part of the electrode unit system, and the transparent column were then covered with dental acrylic.

**Recording and data acquisition.** The electrical signals from the mouse brain were relayed through a custom-made recording system to a computer, located outside the recording room. Briefly, 1–1.5 m long cable was constructed using eight intertwined bare copper wires (single 8058-Magnet Wire) and placed inside a PVC tubing. The cable was soldered accordingly to an LT1112S8#PBF General Purpose Amplifier 2 Circuit 8-SO (Digi-Key) on one end and to an MMA25-011 connector plug with male pins (Digi-Key) to the other end. The connector plug end of the cable was attached to a 10-channel slip ring commutator (Campden Instruments). Both ends of the cable were covered with J-B Weld™ two-part epoxy. The commutator was joined to a custom-made connector box that also contained two 9 V batteries providing the power supply to the preamplifier. The signal was then transmitted to Model 3600 16-channel amplifier (A-M Systems) (LP: 300 Hz, HP: 0.3 Hz, Gain:1 K). The output signal was led to a USB 6009 14 bit A/D converter (National Instruments) connected to a laptop computer. Igor 8.0 (Wavemetrics) NIDAQ tools were used to record the two channels in each mouse at a sampling rate of 2048/s⁻¹. During the recording sessions (pre-stroke ∼21 days after stroke), the mice were housed in clear polycarbonate buckets and remained connected to the recording cable. Mouse motion was captured by an infrared-sensitive USB camera (11–13 frames/s) with iSpy software.

**Rehabilitation.** For rehabilitation during EEG recording, a rehabilitation box with an open top lid was used because the headset and the recording cable prevented the mouse from entering the box. Prior to pre-stroke recording, all mice underwent training to exclusively reach with their right forelimb. Only mice exhibiting exclusive right-forelimb reaching were included in the study. Mice were carefully monitored to ensure that rehabilitated mice continued to use only their right forelimb during post-stroke rehabilitation. Rehabilitation started 10 days after the stroke and continued 5 days a week. EEG recordings were performed on non-rehabilitation days.

**Analysis of LFP recording.** All data analyses were carried out using custom-written procedures in IgorPro 8.0 (Wavemetrics) using its built-in functions for root mean squared (RMS) measurement, FIR filtering, fast Fourier transform (FFT), Hilbert amplitude, etc. First, using a FIR filter [using at least 401 coefficients or with the number of coefficients determined as = int(50/(22*(HF-LF)/SR)), where int is the integer part, 50 is the cut-off of the filter in dB, HF and LF are the high and low frequencies, respectively, of the bandpass and SR is the sampling rate (2048 s⁻¹)], we bandpass filtered the raw recordings, at 0.5–4 Hz for δ, 5–12 Hz for θ and 30–120 Hz for γ oscillations. This digital FIR filtering did not change the phase of the oscillations. The bandpass-filtered traces were then used for calculating the RMS values for vigilant state determination.

The RMS values for the γ and δ bandpass filtered recordings were calculated in 8 s long epochs shifted by 1 s. The RMS(γ)/RMS(δ) ratios were calculated on a point-by-point basis and plotted for 12 h recording periods corresponding to the light or the dark cycles. A histogram of

the RMS(γ)/RMS(δ) ratios was plotted for an entire 12 h (43 200 s) recording period. The histograms were the best fit by two Gaussian distributions. The Gaussian with the lowest mean and variance, indicating a low RMS(γ)/RMS(δ) ratio, was considered as the starting point for identifying NREM periods. To further refine the NREM periods, we used the probabilities relative to the SD of a normal distribution. In such distributions, the probability of values lying between the mean and $1 \times SD$ from the mean is 68.27%, between the mean and $2 \times SD$ from the mean is 95.45% and between the mean and $3 \times SD$ from the mean is 99.73%. Thus, for each 8 s segment of the RMS(γ)/RMS(δ) ratios, we calculated the probability of the given segment belonging to the Gaussian with the lowest mean in the distribution of RMS(γ)/RMS(δ) ratios. The decision about a given segment belonging (or not) to NREM was made on the basis of averaging probabilities in 8 s segments both before and after the 1 s segment of the RMS(γ)/RMS(δ) ratio. A continuous probability value as a function of time was calculated using three different probability levels assigned point-by-point (1 s epochs) for values of the RMS(γ)/RMS(δ) ratios (G/D) in the lowest mean Gaussian distribution with a standard deviation of SD as fitted using the Igor Pro 8 Gauss curve-fitting procedure. The probabilities for the 1 s G/D segments were assigned as follows: a value of 1, if $G/D \leq mean + 1 \times SD$; a value of 0.397 [i.e. (95.45 − 68.27)/68.27], if $mean + SD < G/D \leq mean + 2 \times SD$ and a value of 0.067 [i.e. (99.73 − 95.45)/68.27], if $mean + 2 \times SD < G/D \leq mean + 3 \times SD$. From this probability function, a binary NREM classification was constructed by assigning a value of 0 if the average of the probability values was <0.5 during both the 8 s prior and 8 s after the 1 s epoch in question. If the average of the probabilities during the two 8 s segments was ≥0.5, the value assigned to the binary NREM classifier was 1. Once the NREM periods were thus identified, we calculated the RMS(θ)/RMS(δ) ratios of the θ and δ bandwidth-filtered recordings of the segments lying outside the already identified NREM periods. These were the segments to be subsequently classified as REM or AWAKE, as follows. First, we constructed all-point histograms of the RMS(θ)/RMS(δ) ratios of these segments, as θ/δ ratios are known to be highest during REM sleep in rodents. These histograms could be best fitted by two Gaussians. We considered the intersection of the two Gaussian distributions (the point where the probabilities of belonging to either of the distributions are equal) to define the objective threshold for the RMS(θ)/RMS(δ) ratios separating the two distributions. The initial classification between REM and AWAKE was done based on the two RMS(θ)/RMS(δ) distributions. The 1 s epochs were classified as REM if their RMS(θ)/RMS(δ) was above the objective threshold (i.e. it fell into the Gaussian with the larger mean), and if the segment was flanked at least on one side by an already identified NREM period. The activity (Act) levels of the mouse as obtained from the frame-by-frame subtraction of the video recordings were used to further refine the distinction between REM and AWAKE. The Act was analyzed separately during the established NREM periods, when the animal was resting, and separately during REM or AWAKE periods (outside of the NREM periods). The REM classified segments based on the RMS(θ)/RMS(δ) distributions were reclassified as follows: where the Act was greater than the median Act (during REM or AWAKE), the REM segments were reclassified as AWAKE and original AWAKE segments where the Act was lower than the median Act (during NREM) were reclassified as REM. Where transitions of AWAKE to REM occurred, the REM was also reclassified as AWAKE. The magnitude of the spectrum power was calculated using FFT across the entire data segment. Frequency bands were created by averaging the spectral values in the 1–4 Hz (delta), 4–12 Hz (theta), 30 −57 Hz (low gamma) and 63–119 Hz (high gamma) ranges. Spectrum powers in each frequency band were normalized to the pre-stroke baseline value.

## Human electroencephalogram

**Subject.** Individuals with stroke aged 18 years or older were recruited from the inpatient rehabilitation facility at the University of California, Irvine Medical Center (20 Male, 7 Female; 19 White, 3 Asian control and 16 White, 8 Asian, 3 Black stroke patients 2 Hispanic, 25 Non Hispanic control and 2 Hispanic, 25 Non Hispanic; Age 49–67 years [median 58]). An EEG recording was obtained around the time of IRF admission, IRF discharge, two time between as possible, and around day 90 after stroke onset. Approval was granted by the IRB at UC Irvine, where the reference number was 2004–3852. All the data from human participants were collected at University of California, Irvine, where the ethics approval was granted. All participants provided informed written consent.

**Electroencephalography Acquisition and Preprocessing.** A three-minute resting-state EEG recording was obtained from each participant using a dense-array 256-lead Hydrocel net (Electrical Geodesics Inc.). EEG data were sampled at 1,000 Hz using a high input impedance Net Amp 300 amplifier and Net Station 4.5.3 software (Electrical Geodesics Inc.). Raw and unfiltered EEG data were imported to Matlab for the following offline preprocessing steps: [1] removal of 64 leads from the cheek and neck region; [2] re-referencing to the average signal across the remaining 192 leads; [3] 50 Hz low-pass filtering; [4] segmenting the data into 1-s nonoverlapping epochs with detrending; and [5] muscle artifact removal during visual inspection. Ocular and cardiac artifacts were removed using an Infomax independent components analysis (ICA) in EEGLAB[126] prior to an additional round of visual inspection after transforming the data to electrode space in order to assess ICA accuracy.

**Electroencephalography Measurements.** Spectral analysis of the data using a discrete Fast Fourier transform allowed for the computation of power and coherence at all 192 leads, from 1 to 40 Hz. Measures of relative power in the gamma band (30–40 Hz) for each electrode were obtained by dividing power in that band by the total power summed over the entire 1–40 Hz range.

## AUT00201 pharmacokinetics

Electrophysiological recordings were performed from HEK cells stably expressing hKv3.1b. Cells were maintained in minimum essential medium (Sigma) with 10% fetal bovine serum (Sigma), 2 mM L-glutamine (Sigma), 10 ml/l penicillin-streptomycin (Sigma) and 0.5 mg/ml geneticin (Biowest) and maintained in a 5% $CO_2$ incubator at 37 °C. HEK cells were grown on coverslips 18–24 h preceding recordings and transferred to extracellular solution (137 mM NaCl, 1.8 mM $CaCl_2$, 1 mM $MgCl_2$ 4 mM KCl, 10 mM HEPES, and 10 mM glucose, pH 7.4) Recordings were made in the whole cell configuration, using a Multiclamp 700B amplifier (Axon instruments, Foster City, CA). The patch electrodes had a resistance of 2–3 MΩ when filled with intracellular solution (110 mM KCl, 0.2 mM EGTA, and 40 mM HEPES, 1 mM $MgCl_2$, 0.1 mM $CaCl_2$, 5 mM Na2phosphocreatine, pH 7.2). All data were low pass filtered at 5 kHz and digitized using Digidata 1440 A interface (Molecular Devices LLC, Sunnyvale, CA, USA). Data were sampled at 50 kHz. Data were analyzed using pCLAMP10 software. Average data are expressed as means +/− sem. Conductance values were obtained by dividing current by the electrochemical driving force (IK / (Vm-Ek)). Normalized conductance-voltage plots were obtained by normalizing conductance (G) to maximal conductance (Gmax) and fit using the Boltzmann isoform G = Gmax / [1 + exp ((V - V1/2) / k)], where V1/2 is the voltage at half-maximal activation and k is the slope factor. All the experiments were carried out in paired mode: control data (0.1% DMSO) vs treatment (AUT00201). Cell membrane potentials were held at −70 mV and currents were evoked by voltage steps (200 ms duration) from −90 mV to +40 mV (in 10 mV increments). Vehicle (0.1% DMSO) or AUT00201 were applied for at least 3 min prior to the subsequent voltage step protocol application. Drugs were applied to the bath by a perfusion pump (flux rate 1 ml/min). AUT00201 (0.02 and 0.2 uM in DMSO) were dissolved from stock solutions and diluted to the final

concentration on the day of the experiment. The highest final concentration of DMSO used was 0.1%.

## DDL-920 pharmacokinetics
Mice were administered DDL-920 via pipette feeding at a dose of 10 mg/kg. DDL-920 was solubilized in water and then condense milk was added to it (Water: Condense = 40:60). Mice were euthanized and perfused for 1 h and brain tissue was collected. No notable changes in mice behavior that would be indicative of drug-induced toxicity were observed. Analysis of brain concentrations was done at the UCLA Pasarow Mass Spectrometry Lab (PMSL; Julian P. Whitelegge, Ph.D., Director). Tissue samples were homogenized in a bead beater using 5 volumes of ice-cold 80% acetonitrile (1/5; mg of tissue/ μl of 80% ACN). Solutions were clarified by centrifugation (16,000 x g, 5 min) and the supernatants were transferred to new tubes and lyophilized. Samples were reconstituted in 100 μl of 50/50/0.1 (Water/Acetonitrile/Formic Acid) prior to analysis via liquid chromatography-tandem mass spectrometry (LC-MS/MS). A targeted liquid chromatography-tandem mass spectrometry (LC-MS/MS) assay was developed for each compound on an Orbitrap LTQ XL (Thermo Fisher Scientific) coupled to a Dionex Ultimate 3000 HPLC system (Thermo Fisher Scientific) with a Phenomenex analytical column (Kinetex 1.7 μm C18 100 Å 100 ×2.1 mm). The HPLC method utilized a mixture of solvent A (99.9/1 Water/Formic Acid) and solvent B (99.9/1 Acetonitrile/Formic Acid) and a gradient was used for the elution of the compounds (min/%B: 0/ 20, 3/20, 19/99, 20/99, 21/20, 30/20). In this assay, the detection of fragmented ions originating from each compound at specific LC retention times was utilized to ensure specificity and accurate quantification in the complex biological samples.

An internal standard (IS) was added to every sample to account for compound loss during sample processing. Standards were made in drug naïve brain lysates with increasing amounts of DDL-920 (S1, S2: 0 pmol/ S3, S4: 1 pmol/ S5, S6: 10 pmol/ S7, S8: 100 pmol, S9, S10: 1000 pmol). The standard curve was made by plotting the amount of compound (pmol) per standard vs. the ratio of measured chromatographic peak areas corresponding to that of each analyte over that of the IS (analyte/IS). The trendline equation was then used to calculate the absolute concentrations of each compound in brain tissue.

## Statistical analyses
Mice were randomly allocated to treatment conditions, and all results were analyzed with the investigator blinded to treatment conditions. All data are reported as mean ± SEM. The normal distributions of all the data were tested by Shapiro-Wilk test. Parametric tests were used for normally distributed datasets, and nonparametric tests were applied to data not normally distributed. For comparisons of two groups, paired and unpaired t tests or Wilcoxon tests were used. For comparisons of more than three groups, one-way ANOVA tests followed by Tukey's multiple comparison test or Kruskal-Wallis test followed by Dunn's multiple comparison test were used. For comparisons of data across time points, two-way repeated measure ANOVA followed Sidak's multiple comparison test or mixed-effects model followed by Tukey's multiple comparison test were used. For categorical data, the Chi-squared test followed by Bonferroni test was used. For correlation analysis, Pearson correlation or Spearman correlation were used. A statistical significance was defined as a P value < 0.05. A generalized linear mixed model was used for response probability in the optogenetic patch-clamp experiment. Statistical analyses were conducted using Graph Pad Prism (Graphpad Software Inc.) and R (R Development Core Team).

## Reporting summary
Further information on research design is available in the Nature Portfolio Reporting Summary linked to this article.

## Data availability
All data generated in this study are provided in the main text, the Supplementary materials, or Source Data. Raw imaging or other datasets from this paper will be made available upon request to the corresponding author, NO and STC. Source data are provided with this paper.

## Code availability
Custom code developed in this study has been deposited in Zenodo and is available at https://doi.org/10.5281/zenodo.14838078.

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

## Acknowledgements
NO is supported by JSPS KAKENHI Grant (Fostering Joint International Research:18KK0276) and Uehara Memorial Foundation. STC is supported by Dr. Miriam and Sheldon G. Adelson Medical Research Foundation. We thank Dr. Dario Ringach and Dr. Joshua Trachtenberg for technical assistance in 2 photon microscopy, Dr. William Zeiger and Dr. Campos Baruc for discussion about chemogenetics, Dr. Guido Faas for 3D printing materials, Jaehyok Jeon for behavioral analysis, and the members of S.T.C.'s laboratory for helpful discussions. We thank Addgene and their depositors for providing virus constructs. Schematic diagrams were created with Biorender.com.

## Author contributions
Conceptualization: N.O., S.T.C. Methodology: N.O., X.W., J.C., J.V., S.C.C., I.M., N.P., A.M., G.A., M.G. Investigation: N.O., X.W., M.H., W.D., S.A., F.A., J.B., F.Q., Visualization: N.O. Funding acquisition: N.O., S.T.C. Project administration: N.O., S.T.C., S.C.C., M.G. Supervision: N.O., S.T.C., S.C.C., I.M. Writing – original draft: N.O., S.T.C.. Writing – review & editing: N.O., S.T.C., S.C.C., I.M., M.G., X.W.

## Competing interests
A.M., N.P., G.A., and M.G. are employees of and shareholders in Autifony Therapeutics Ltd. The other authors declare no competing interests.

## Additional information

Naohiko Okabe [1,8] ✉, Xiaofei Wei [2], Farah Abumeri[1], Jonathan Batac[1], Mary Hovanesyan[1], Weiye Dai [1],
Srbui Azarapetian[1], Jesus Campagna[3], Nadia Pilati[4], Agostino Marasco [4], Giuseppe Alvaro[4], Martin J. Gunthorpe [5],
John Varghese[3], Steven C. Cramer[6], Istvan Mody [1,7] & S. Thomas Carmichael [1,8] ✉

[1]Department of Neurology, David Geffen School of Medicine, UCLA, Los Angeles, CA 90095, USA. [2]Department of Neurosurgery, David Geffen School of
Medicine, UCLA, Los Angeles, CA 90095, USA. [3]The Drug Discovery Lab, Mary S. Easton Center for Alzheimer's Disease Research, Department of Neurology,
David Geffen School of Medicine, UCLA, Los Angeles, CA 90095, USA. [4]Autifony Srl, Istituto di Ricerca Pediatrica Citta' della Speranza, Via Corso Stati Uniti,
4f, 35127 Padua, Italy. [5]Autifony Therapeutics Limited, Stevenage Bioscience Catalyst, Stevenage SG1 2FX, UK. [6]Department of Neurology, UCLA, California
Rehabilitation Institute, Los Angeles, CA 90095, USA. [7]Department of Physiology, David Geffen School of Medicine, UCLA, Los Angeles, CA 90095, USA.
[8]These authors jointly supervised this work: Naohiko Okabe, S. Thomas Carmichael. ✉e-mail: Nokabe@mednet.ucla.edu; SCarmichael@mednet.ucla.edu

