## [Transparent Peer Review file · Nature Communications]

Parvalbumin interneurons regulate rehabilitation-induced functional recovery after stroke and identify a rehabilitation drug

Corresponding Author: Dr Naohiko Okabe

Version 0:

Reviewer comments:

Reviewer #3

(Remarks to the Author)

The study by Okabe et al. explores the neural mechanisms underlying motor recovery following stroke, a condition currently lacking effective medical therapies. Their findings demonstrate that rehabilitation enhances synapse formation within specific motor cortex circuits, particularly between presynaptic parvalbumin interneurons and postsynaptic stroke-projecting neurons. Rehabilitation strengthens motor performance by improving functional connectivity, which diminishes when stroke-projecting neurons are inhibited. Structural and functional changes in stroke-projecting neurons were observed, including reduced dendritic spine density and altered synaptic inputs. Furthermore, the activation of parvalbumin interneurons—critical for generating gamma oscillations—was shown to be essential for motor recovery. The pharmacological enhancement of parvalbumin interneuron function replicated the effects of rehabilitation, presenting a promising direction for molecular-based therapies to complement rehabilitation.

The authors addressed some critical comments. However, concerns remain regarding specific experimental results, particularly those derived from patch-clamp analyses in Figure 3. Statistical limitations are notable, with several groups having a sample size of $n=5$ (e.g., Resting Vm-Stroke; Rin-Sham; Rin-Stroke+Rehab), resulting in high data variance that undermines statistical robustness. Consequently, conclusions about the absence of changes in intrinsic excitability are unsupported by robust evidence and should be reconsidered. Additional parameters such as rheobase, action potential threshold, and I/F curves should be analyzed to provide a more comprehensive evaluation of intrinsic excitability. Claims regarding no change in intrinsic excitability should be omitted unless supported by more rigorous statistical evidence.

On page 12, the reported increase in EPSC amplitude without changes in phasic current is explained by a potential decrease in EPSC frequency; however, Figure 3e does not support this claim, as no statistically significant reduction is evident. Moreover, the authors' conclusion that stroke animals receive higher inhibitory inputs without substantial excitatory input changes contradicts the significant increase in EPSC amplitude depicted in Figure 3f.

For IPSCs, while a significant increase in IPSC frequency is observed in Stroke animals (Figure 3i), no such increase is evident in the Stroke+Rehab group, contrary to the authors' claim. If this discrepancy arises from a figure labeling error, it should be corrected, and significance levels clearly indicated.

Based on the aforementioned inconsistencies in data analysis or presentation, a Summary Table with all statistical analysis results indicating the number of animals and the number of cells would be required to support the authors' claims.

Reviewer #4

(Remarks to the Author)

This is a very interesting paper that I believe will impact the way the field thinks about stroke and recovery. The authors largely addressed my prior comments. I appreciate that the authors have added additional analyses on the electrophysiological recordings and modified the discussion to tone down some of the interpretations where other mechanisms may be contributing factors. They also address some of the inconsistencies noted in my prior review. I have one minor request that I believe will help people reading this paper. That is, for fig 1O, it would be very helpful to indicate

which animals are from the control, stroke and stroke + rehab groups. I get the point of performing the correlation with all animals, but distinguishing the different groups either by color coding or some other means would help the reader better understand which data points were contributed by which group.

Thank you for your feedback and constructive comments. We acknowledge that the previous whole-cell patch clamp recordings may not be ideal for assessing intrinsic excitability due to inherent methodological limitations. Consequently, following Reviewer 3's suggestion, we have omitted claims about intrinsic excitability changes. To enhance clarity and accuracy, we have reformatted the patch-clamp figures incorporating the rise-rate analysis, omitted data descriptions without statistical significance, and restructured the manuscript for better readability. Additionally, we created a comprehensive statistical summary table detailing all analysis results across the study, demonstrating the robustness of our dataset, which includes 649 mice and 27 human patients. These data consistently support our conclusion on the pivotal role of stroke-projecting neuron-PV interneuron circuits in rehabilitation-induced stroke recovery.

Reviewer #3:

The authors addressed some critical comments. However, concerns remain regarding specific experimental results, particularly those derived from patch-clamp analyses in Figure 3. Statistical limitations are notable, with several groups having a sample size of $n=5$ (e.g., Resting Vm-Stroke; Rin-Sham; Rin-Stroke+Rehab), resulting in high data variance that undermines statistical robustness. Consequently, conclusions about the absence of changes in intrinsic excitability are unsupported by robust evidence and should be reconsidered. Additional parameters such as rheobase, action potential threshold, and I/F curves should be analyzed to provide a more comprehensive evaluation of intrinsic excitability. Claims regarding no change in intrinsic excitability should be omitted unless supported by more rigorous statistical evidence.

We would like to thank the reviewer for the insightful comments about our claims regarding the intrinsic properties of the neurons in our study. We agree that the number of cells was not adequate to draw meaningful conclusions about these parameters. However, we also should recognize that whole-cell patch clamp recordings in brain slices are not the best approach for measuring intrinsic excitability changes of neurons. The pipette solution used for the whole-cell recordings inadvertently alters the intracellular milieu of the cells. To avoid this, cell-attached recordings are the preferred method for determining the characteristic properties of neurons (Perkins 2008, PMID: 16554092). Optical methods for finding a meaningful Vm value have also been suggested (Lazzari-Dean et al., 2021 PMID: 33651949). In addition, there is another aspect of patch clamp recordings in brain slices that is rarely considered. Such recordings are inherently biased for the healthiest neurons found in the slices, as no gigaseal patches can be established on the membranes of cells injured in some way. This bias is even more pronounced when one is sampling cells from a brain area that has been compromised in some way, in our case, by a stroke to the same hemisphere. Considering all the bias in gathering information about intrinsic cell properties using whole-cell patch clamp recordings in slices, we have heeded the advice of the reviewer that "claims regarding no change in intrinsic excitability should be omitted". In future studies, we will strive to use better approaches to measure the passive and active properties of the neurons in our preparations. Instead, we have focused on new and more thorough analyses of the synaptic events recorded in slices.

On page 12, the reported increase in EPSC amplitude without changes in phasic current is explained by a potential decrease in EPSC frequency; however, Figure 3e does not support this

claim, as no statistically significant reduction is evident. Moreover, the authors' conclusion that stroke animals receive higher inhibitory inputs without substantial excitatory input changes contradicts the significant increase in EPSC amplitude depicted in Figure 3f.

For IPSCs, while a significant increase in IPSC frequency is observed in Stroke animals (Figure 3i), no such increase is evident in the Stroke+Rehab group, contrary to the authors' claim. If this discrepancy arises from a figure labeling error, it should be corrected, and significance levels clearly indicated.

Since our patch-clamp experiments showed distinctive effects in frequency and amplitude analyses, the data on mean phasic current (additionally influenced by the decay time of the events) may be more confusing than informative. To streamline the manuscript, we omitted both the mean phasic current data and the descriptions of changes without statistical significance. As a result, our data demonstrated the following key findings:

1. Stroke increases EPSC amplitude compared to Sham.
2. Rehab increases EPSC frequency compared to Stroke.
3. Stroke decreases the reliability of evoked EPSC responses by optogenetic stimulation of thalamic axons.
4. Stroke increases IPSC frequency compared to Sham.
5. Stroke decreases the fraction of fast-rising large amplitude IPSCs, and Rehab recovers these events.

In the EPSC recordings, the EPSC frequency/probability data roughly align with our histological findings—for example, the increased EPSC frequency after rehabilitation corresponds to spine density recovery, and the decreased reliability of evoked EPSC responses to optogenetic stimulation corresponds to reduced synaptic inputs from the thalamus. These results suggest that structural synaptic changes are more closely related to EPSC frequency than EPSC amplitude. As discussed, EPSC amplitude changes are likely influenced by multiple factors, including postsynaptic receptor expression, presynaptic vesicle release probability, and phasic/tonic inhibitory inputs mediated by transcriptional regulation. Intrinsic excitability should be one of these factors. Given the complexity of these mechanisms, future studies should employ systematic approaches, such as transcriptome analysis combined with whole-cell and cell-attached patch-clamp recordings, and optical approaches for probing membrane potential changes.

In the IPSC recordings, stroke animals exhibited an increased IPSC frequency, with no significant difference observed between Stroke and Rehab animals. Rise-rate analysis revealed a reduction in the fraction of fast-rising large amplitude IPSCs, presumably generated by PV interneurons (Chen et al., 2018, PMID: 30105300) in stroke animals, which was restored by rehabilitation, consistent with our structural analysis of PV synapses, linking structural and functional synaptic plasticity in stroke-projecting/PV interneuron circuits. Just like for EPSC recordings, future studies should investigate the mechanisms underlying IPSC changes using systematic approaches.

We reformatted the patch-clamp data figures to include the rise-rate analysis (Fig. 4). We restructured the manuscript by positioning the patch-clamp data after the GRASP analysis for better readability (Pages 14-15).

Based on the aforementioned inconsistencies in data analysis or presentation, a Summary Table with all statistical analysis results indicating the number of animals and the number of cells would be required to support the authors' claims.

We appreciate the reviewer's feedback regarding the statistical summary table. In response, we have created a comprehensive summary table detailing all statistical analysis results across the study. Our dataset includes data from 649 mice and 27 human patients, encompassing behavioral tests, histological analyses, 2-photon calcium imaging, EEG, and patch-clamp recordings combined with chemogenetic, optogenetic and pharmacological manipulations. These results consistently highlight the critical role of neuronal circuits formed by stroke-projecting neurons and PV interneurons in rehabilitation-induced post-stroke recovery. We believe our dataset is substantial, robust, and comprehensive, providing strong support for our main conclusions.

Reviewer #4:

I have one minor request that I believe will help people reading this paper. That is, for fig 1O, it would be very helpful to indicate which animals are from the control, stroke and stroke + rehab groups. I get the point of performing the correlation with all animals, but distinguishing the different groups either by color coding or some other means would help the reader better understand which data points were contributed by which group.

Thank you for your comments regarding the figure. We edited the figure to distinguish the groups.